

# A new parrot taxon from the Yucatán Peninsula, Mexico—its position within genus *Amazona* based on morphology and molecular phylogeny

Tony Silva[1], Antonio Guzmán[2], Adam D. Urantówka[3] and Paweł Mackiewicz[4]

[1] IFAS/TREC Advisory Committee, University of Florida, Miami, FL, United States of America
[2] Laboratorio de Ornitología, Facultad de Ciencias Biológicas, Universidad Autónoma de Nuevo León, Nuevo León, Mexico
[3] Department of Genetics, Wroclaw University of Environmental and Life Sciences, Wroclaw, Poland
[4] Faculty of Biotechnology, University of Wrocław, Wrocław, Poland

## ABSTRACT

Parrots (Psittaciformes) are a diverse group of birds which need urgent protection. However, many taxa from this order have an unresolved status, which makes their conservation difficult. One species-rich parrot genus is *Amazona*, which is widely distributed in the New World. Here we describe a new *Amazona* form, which is endemic to the Yucatán Peninsula. This parrot is clearly separable from other *Amazona* species in eleven morphometric characters as well as call and behavior. The clear differences in these features imply that the parrot most likely represents a new species. In contrast to this, the phylogenetic tree based on mitochondrial markers shows that this parrot groups with strong support within *A. albifrons* from Central America, which would suggest that it is a subspecies of *A. albifrons*. However, taken together tree topology tests and morphometric analyses, we can conclude that the new parrot represents a recently evolving species, whose taxonomic status should be further confirmed. This lineage diverged from its closest relative about 120,000 years ago and was subjected to accelerated morphological and behavioral changes like some other representatives of the genus *Amazona*. Our phylogenies, which are so far the most comprehensive for *Amazona* taxa enabled us to consider the most feasible scenarios about parrot colonization of the Greater and Lesser Antilles and Central America from South America mainland. The molecular dating of these migrations and diversification rate were correlated with climatic and geological events in the last five million years, giving an interesting insight into Amazon parrot phylogeography and their evolution in general.

# INTRODUCTION

## The genus *Amazona* and its taxonomic changes

*Amazona* (Amazon parrots) is the most species-rich genus within the Androglossini tribe (*Schodde et al., 2013*). The Amazon parrots are strictly neotropical with a distribution that extends from northern Mexico through Mesoamerica and the Caribbean to much

Corresponding author
Tony Silva,
Antonio.Silva@vecenergy.com

of South America, with the southernmost distribution reaching the provinces of Santa Fé and Córdoba in Argentina (*Darrieu, 1983*). They are characterized by medium to large size, strong-heavy bill, short-rounded tail, prominent naked cere and a distinct notch in the upper mandible (*Forshaw, 1973*; *Juniper & Parr, 1998*). Their body plumage is predominantly green with variable colorations on the head, breast, shoulders, and/or flight feathers. Red, yellow, white and blue are dominating colors in their head. The tail is squared in shape and often banded with red and blue stripes. The variation of these accenting colors is one of the morphological features commonly used to distinguish the species and subspecies. However, phylogenetic analyses of mitochondrial DNA (mtDNA) have not always supported the current classification of the *Amazona* group (*Eberhard & Bermingham, 2004*).

When Forshaw published the first edition of his *Parrots of the World* (*Forshaw, 1973*), the genus *Amazona* contained 27 species. No one refuted this arrangement until 1981, when the species number increased to 28 with the elevation of *A. rhodocorytha* to species status (*Barrowclough et al., 2016*); it was formerly regarded as a subspecies of *A. dufresniana* (*Forshaw, 1973*). The first substantial change in the taxonomy of this genus was the transfer of *Amazona xanthops* to the new genus *Alipiopsitta* (*Caparroz & Pacheco, 2006*; *Duarte & Caparroz, 1995*), whose distinctness was first noted by the senior author (*Silva, 1991*). These results opened the floodgates for a series of partial (*Eberhard & Bermingham, 2004*; *Ribas et al., 2007*; *Silva, 2014*; *Urantowka, Mackiewicz & Strzala, 2014*) or complete revisions of the genus *Amazona* (*Russello & Amato, 2004*). Many of these changes have elevated *Amazona* subspecies to the species rank, as in the case of *A. oratrix* and *A. auropalliata* (*Clements et al., 2016*; *Gill & Donsker, 2017*), and four new *Amazona* subspecies have been named (*Lousada, 1989*; *Lousada & Howell, 1997*; *Reynolds & Hayes, 2009*).

While the taxonomic changes were ongoing (H. Sick *in litt.* to T. (Silva, 1988), an additional new species, *A. kawallii*, was described (*Grantsau & Camargo, 1989*). Its validity was firstly questioned (*Vuilleumier, LeCroy & Mayr, 1992*) but reaffirmed soon afterwards by other authors (*Collar & Pittman, 1996*; *Martuscelli & Yamashita, 1997*; *Silva, 2015*). Currently, most of the present checklists assume that the genus *Amazona* contains 30 species, e.g., *Clements et al. (2016)*.

### *Amazona* species native to Mexico and finding the new dimorphic *Amazona*

Mexico is the home of 23 parrot species of which six are endemic (*Gómez Garza, 2014*; *Plasencia-Vazquez & Escalona-Segura, 2014*; *Juniper & Parr, 1998*). Eight of these species belong to the genus *Amazona* and two of them (*Amazona viridigenalis* and *A. finschi*) are found only in Mexico. The Mexican Amazon parrots can be divided into three groups with different coloring: (1) having variable amounts of yellow on the head (*A. oratrix, A. auropalliata*); (2) predominately green with only blue on the head (*A. farinosa guatemalae*); and (3) possessing various tonalities of red in the head invariably accompanied by blue (*Amazona viridigenalis, A. finschi*), yellow (*Amazona xantholora, A. autumnalis*) or white (*Amazona xantholora, A. albifrons*). Monomorphism is the rule in the genus *Amazona*. However, two species, *Amazona albifrons* (all three subspecies) and *Amazona xantholora*,

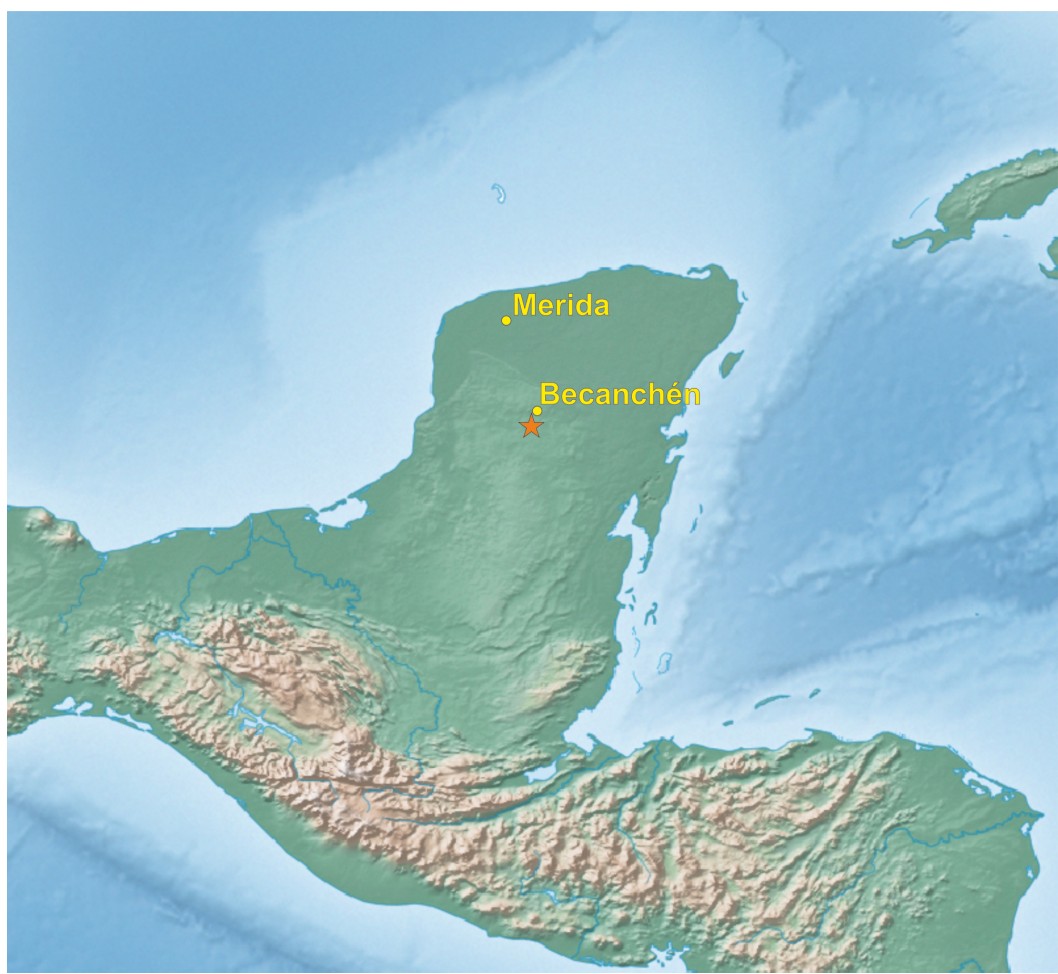

**Figure 1** Map of Yucatán Peninsula with the location of site (asterisk), where the new *Amazona* was found.

display significant dimorphism: males possess much more colorful heads and a more extensively red colored alula and wing speculum (*Gómez Garza, 2014*; *Silva, 1991*).

In the beginning of 2014, during a visit to a remote part of the Yucatán Peninsula, in south of Becanchén in Tekax Municipality (Fig. 1), Miguel A. Gómez Garza sighted parrots with coloration completely different from that of other known species. The birds' appearance and behavior suggested that they belong to the genus *Amazona*. The individuals of this unknown taxon also exhibited sexual dimorphism (Figs. 2 and 3) like the sympatric *Amazona albifrons* and *Amazona xantholora*.

To verify the taxonomic status of the new parrot, we performed a detailed morphological study comparing it with other Mexican *Amazona* species that possess red feathers in the head. Moreover, to establish its phylogenetic position within the genus *Amazona*, we also sequenced three typical mitochondrial markers from the new form and also from *Amazona xantholora*, which had not been previously studied at the molecular level.

This new parrot can be confused with *A. albifrons* and *A. xantholora* in the field when observed at a distance, by their similar size and general appearance. However, its call and

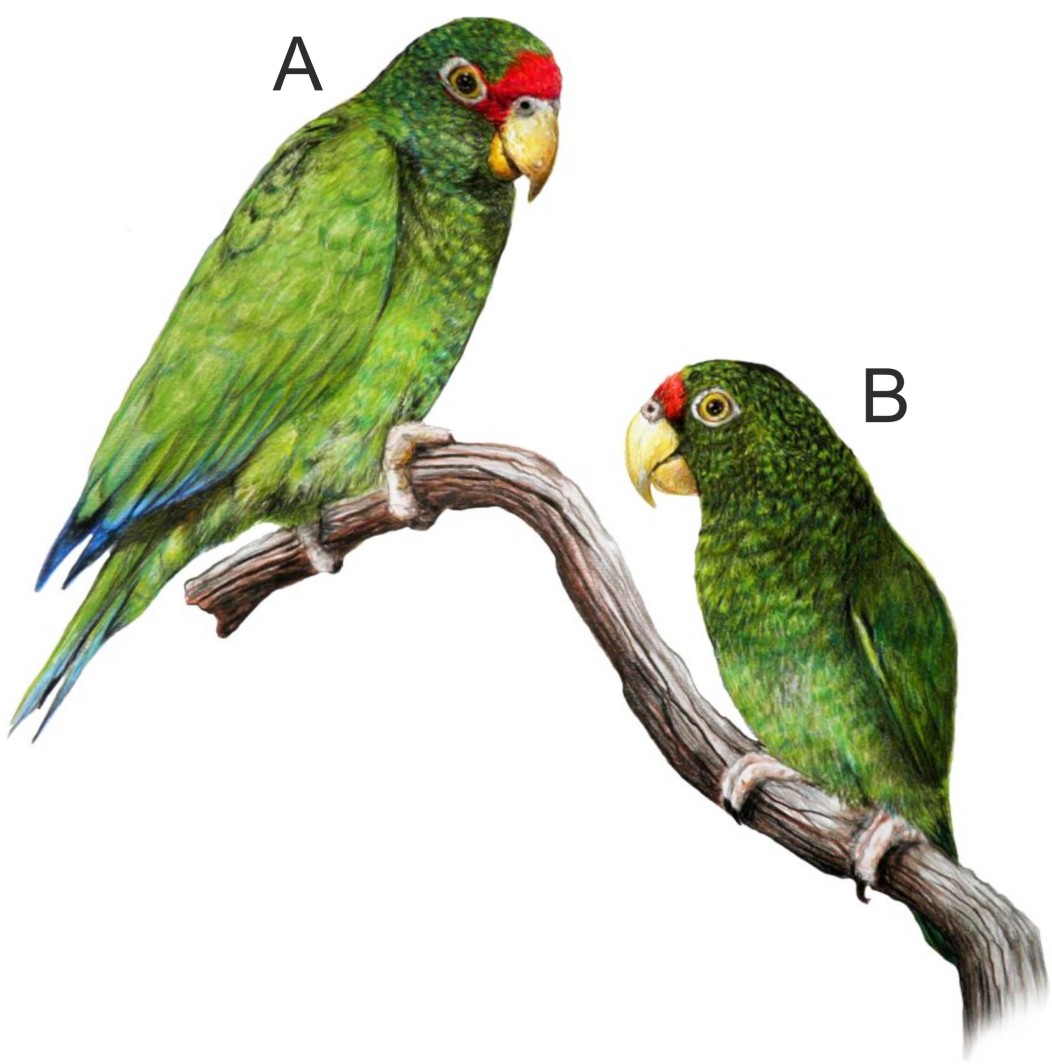

**Figure 2** **Illustration of the new *Amazona*.** Male holotype (A) and female paratype (B). Illustration by Juan García Venegas.

other morphological features are very distinctive and could be used in discrimination of this parrot as a new species, at least under typological, morphological and phenetic species concepts. On the other hand, molecular phylogenetic analyses imply that this parrot could be a subspecies of *A. albifrons*. Therefore, we discussed the pros and cons of these two taxonomic concepts and presented its phylogenetic position in the wide framework of genus *Amazona* evolution and phylogeography.

## MATERIALS AND METHODS

### The new *Amazona* sampling

Living specimens of known morphological types of both sexes (male holotype and female paratype—see Figs. 4–7) of the new *Amazona* were collected in the Yucatán Peninsula in Mexico, south of Becanchén in Tekax Municipality. However, the detailed location is not

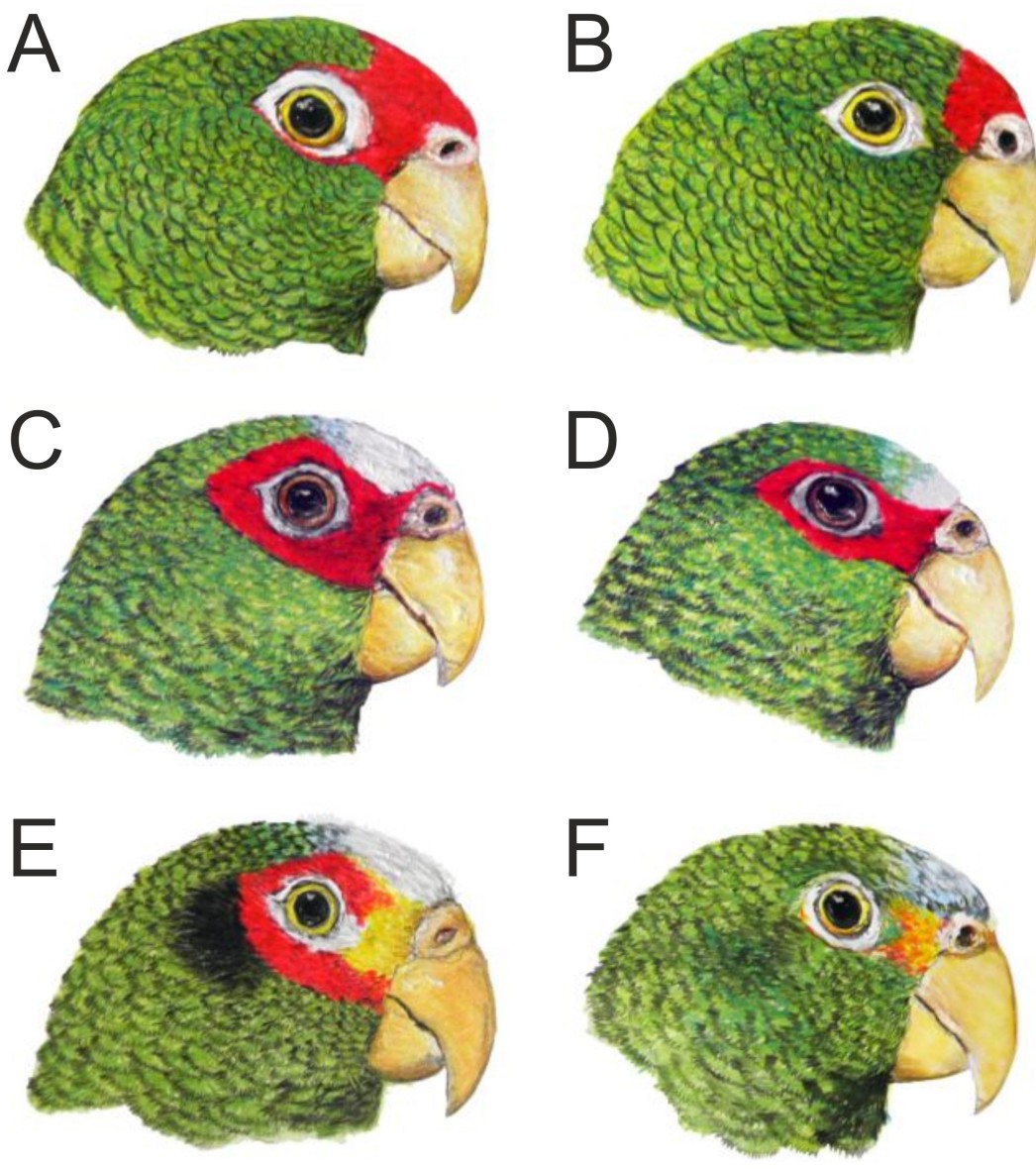

**Figure 3** Head coloration in the male (A) and female (B) of the new *Amazona* in comparison to both sexes of congeners *Amazona albifrons nana* (C, male; D, female) and *Amazona xantholora* (E, male; F, female), also from the Yucatán Peninsula, Mexico. The three taxa are the smallest members of the genus *Amazona*. Illustrations by Juan García Venegas.

provided here due to conservation reasons. Both individuals are now maintained as living birds in Mexico by Miguel Angel Gómez Garza with the permission and authorization of the Procuraduría Federal de Protección al Ambiente (PROFEPA), the national wildlife protection agency. Tail feathers from both specimens were used for DNA isolation and were also deposited in the collection of the Laboratorio de Ornitología, Facultad de Ciencias Biológicas, Universidad Autonóma de Nuevo León, Mexico. This material is assigned the following catalog numbers: MGG01—*Amazona gomezgarzai*—Holotipo—for male feathers and MGG02—*Amazona gomezgarzai*—Alotipo—for female feathers. Both the

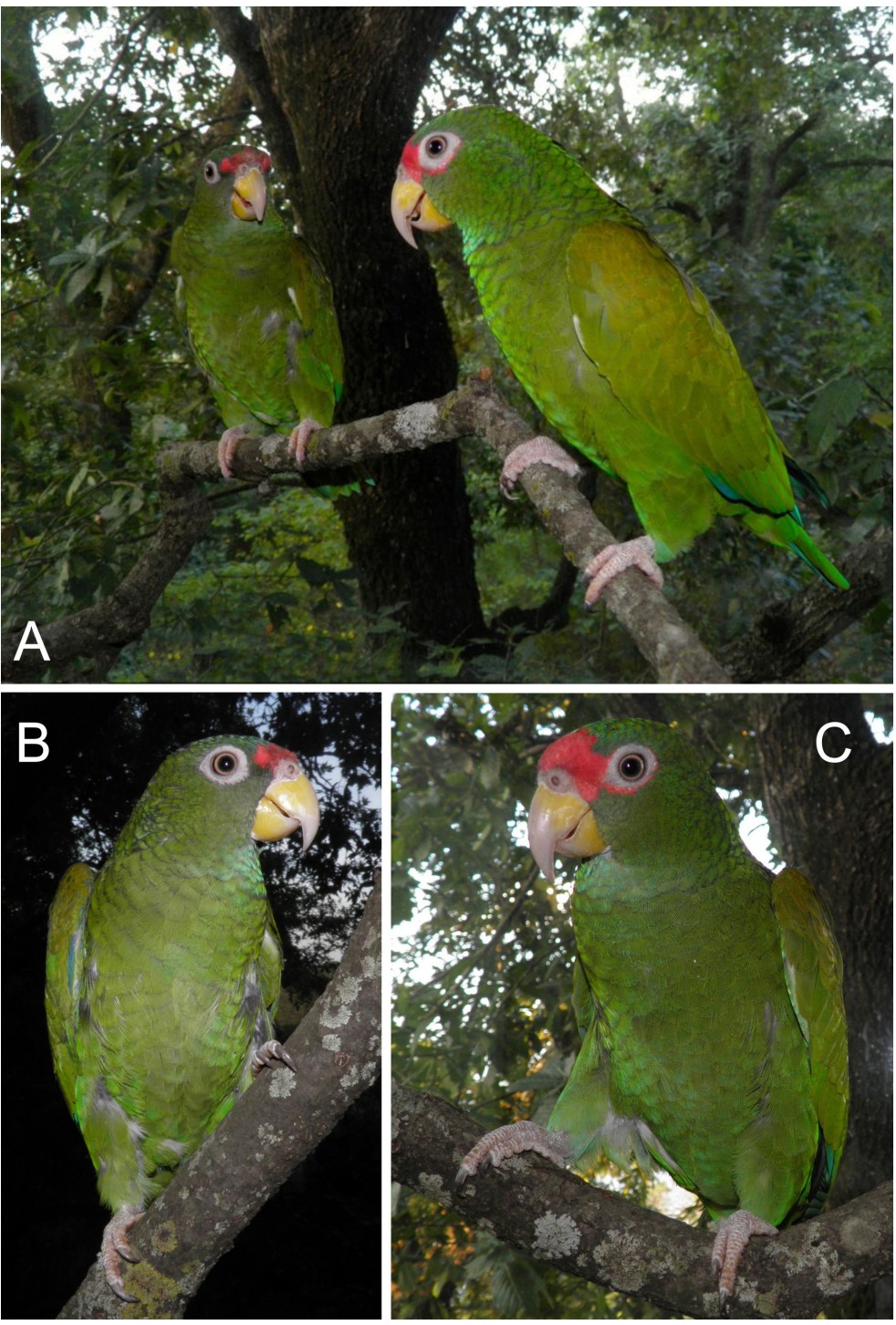

**Figure 4** Photograph of the male holotype (C and A—individual on the right) and female paratype (B and A—individual on the left) of the new *Amazona*.

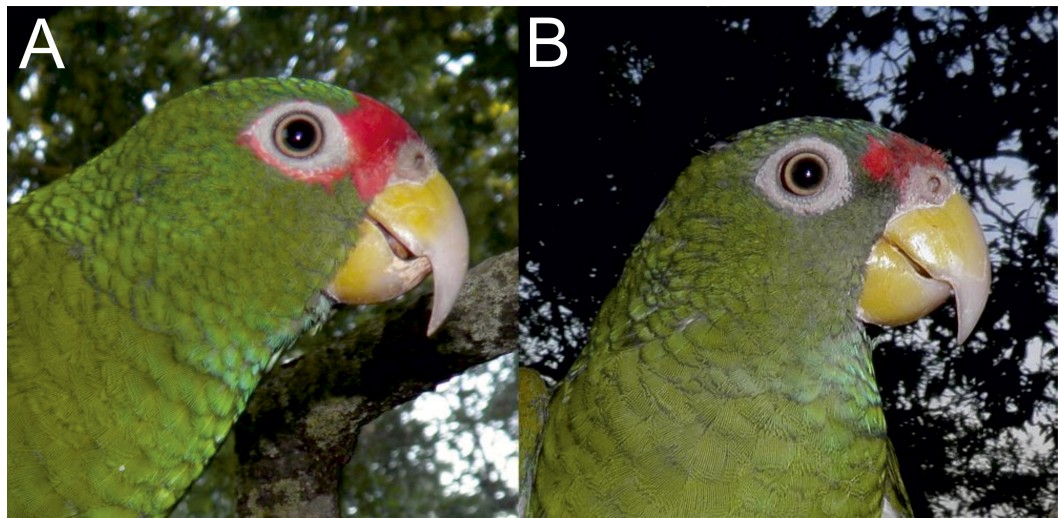

**Figure 5** Photographs of the head of male holotype (A) and female paratype (B) of the new *Amazona*.

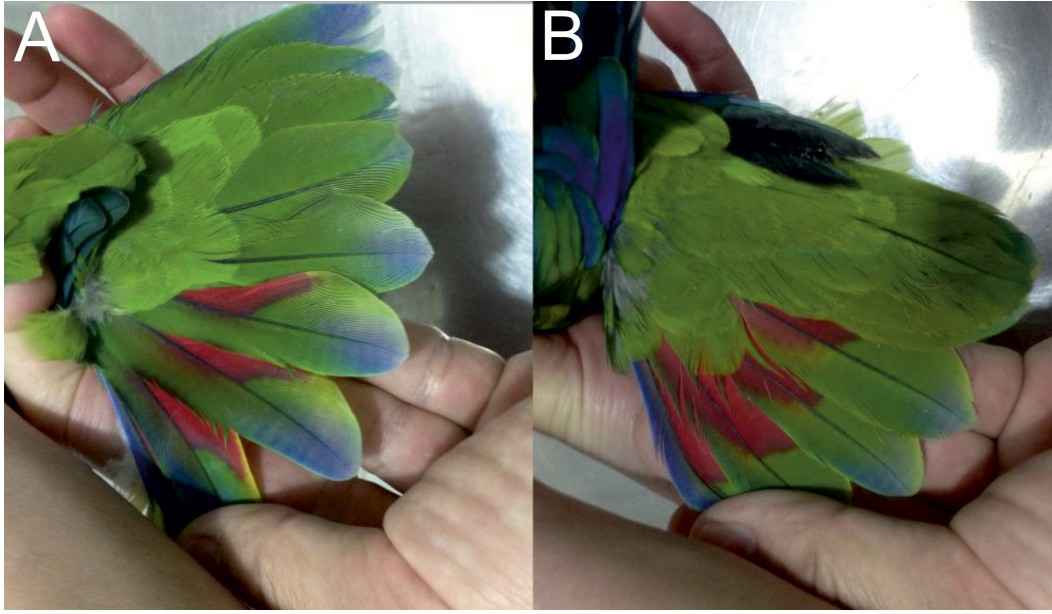

**Figure 6** Open tails showing colored bands of male holotype (A) and female paratype (B) of the new *Amazona*.

living holotype and paratype will be ceded with the authorization of PROFEPA to the Laboratorio de Ornitología, Facultad de Ciencias Biológicas, Universidad Autonóma de Nuevo León, Mexico, upon their death.

We checked the collections of six museums rich in Mexican birds for possibly misidentified specimens of the new taxon that could have been used as type specimens: Museo Nacional de Historia Natural in Madrid (Spain), the collection belonging to the Estación Biológica de Doñana in Seville (Spain), the Field Museum of Natural History

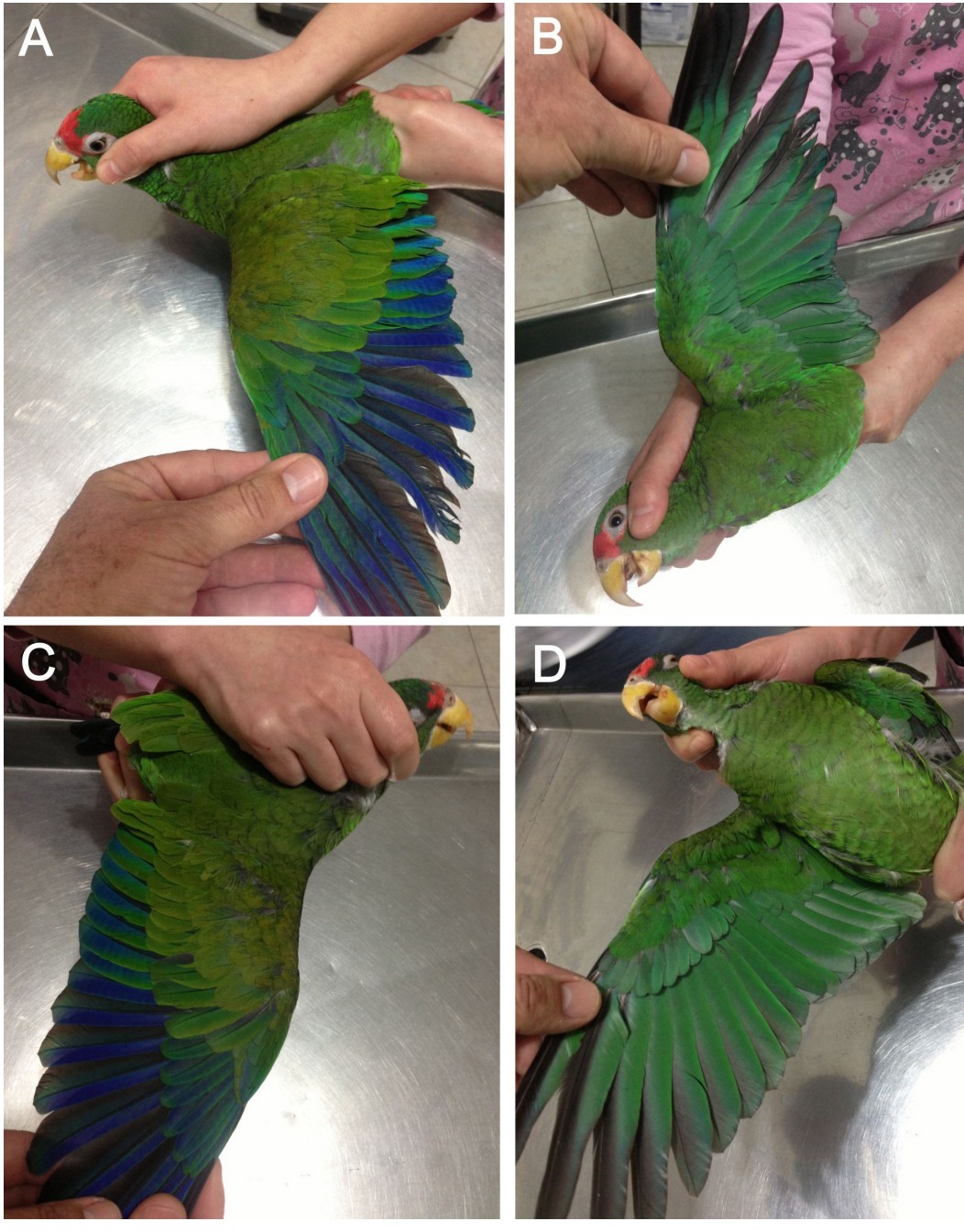

**Figure 7** Open upper (A) and underside of wing (B) of male holotype as well as open upper (C) and underside of wing (D) of female paratype of the new *Amazona*.

in Chicago (USA), Museo de las Aves de México in Saltillo (Mexico), Laboratorio de Ornitología, Universidad Autónoma de Nuevo León (Mexico) and the Laboratorio de Ornitología de la Universidad Nacional Autónoma de México (Mexico). However, we did not find any specimens with characteristics of the new taxon.

The electronic version of this article in Portable Document Format (PDF) will represent a published work according to the International Commission on Zoological Nomenclature (ICZN), and hence the new names contained in the electronic version are effectively published under that Code from the electronic edition alone. This published work and the nomenclatural acts it contains have been registered in ZooBank, the online registration system for the ICZN. The ZooBank LSIDs (Life Science Identifiers) can be resolved and the associated information viewed through any standard web browser by appending the LSID to the prefix http://zoobank.org/. The LSID for this publication is urn:lsid:zoobank.org:act:C4AA8659-8077-4195-9E11-D2EB3635397C. The online version of this work is archived and available from the following digital repositories: PeerJ, PubMed Central and CLOCKSS.

In taxonomic circles, there has been much debate about the deposition of preserved voucher specimens when naming a species (*Donegan, 2008*; *Dubois & Nemésio, 2007*; *Nemesio, 2009*). In the current case, the precarious status of the new *Amazona*, which warrants a listing of Critically Endangered (CR) under IUCN (International Union for the Conservation of Nature) criteria makes the collection of a preserved specimen ethically impossible; the taking of two living specimens will permit their nesting biology in captivity to be studied, as the birds are held in a manner that should allow them to breed; when such breeding takes place, details on incubation and the development of the young can be documented.

A precedent for naming species without the formal deposition of such type was proposed by *Smith et al. (1991)* and was followed by others (*Athreya, 2006*; *Gentile & Snell, 2009*; *Jones et al., 2005*). Three former secretaries of the International Commission on Zoological Nomenclature support the view that the CODE allows the naming of a species without the collection of a voucher specimen in particular circumstances, see *Polaszek et al. (2005)* and *Wakeham-Dawson, Morris & Tubbs (2002)*. Moreover, Article 16.4.2 of the CODE states that where the holotype is an extant individual, a statement of the intent to deposit the individual in a collection upon its death accompanied by a statement indicating the name and location of that collection is sufficient.

Herein, the authors follow *Böhme & Ziegler (1997)* in naming a new species based on a living specimen, but the recommendation by *Dubois (2009)*, who suggested that museums be contacted for the existence of specimens that had not erstwhile been recognized, was followed to no avail for the new *Amazona*. In lieu of an onomatophore specimen of the new parrot, the authors deposited feathers from the holotype and paratype as vouchers in following *Smith et al. (1991)*, per the recommendation of *Donegan (2008)* and in compliance with Article 72.5.1 of the CODE, which states that a type may be an animal or any part of an animal. Article 16.4.2 of the CODE will be met with the deposition in a secure collection of the extant, caged individuals from whom the feathers were removed upon their death. The photographs and illustrations that accompany this description represent the designated holotype and paratype. The authors thus believe that they have met all thresholds to adequately name for the new *Amazona* taxon.

## Morphometric and vocalization data collection and analysis

Adult specimens of parrots were collected evenly throughout the whole year without focusing on a specific season. We observed no great variation of weights between individuals of the same species. We examined them according to five metric features: body weight, total length, length of wing chord, tail length and exposed culmen, as well as six morphological discrete characters: coloration of forehead, lores, cheeks and crown, the presence of black ear patch and black scalloping on contour feathers on the face (Tables 1–3). The measurements were taken by one person (TS) using Fischer Scientific digital caliper with the resolution of 0.1 mm. Three individuals of each sex were measured for each taxon, except for *Amazona autumnalis* in which six birds of unknown sex were taken into account. In the case of the new *Amazona*, only two specimens were collected and analyzed in details because of its endangered status but several small groups with up to 12 individuals were additionally observed and studied in the field. To use the different morphometric features simultaneously in multivariate studies, we normalized their values using the minimum-maximum method: (value−min)/(max–min). The metric features were first averaged per the particular taxon or sex before the normalization. The morphological discrete characters were coded as 1 (when the character was present), 0 (when it was absent) or 0.5 (when it has an intermediate state).

The Principal Component Analysis (PCA) was done using the Statistica software (Version 1.0; StatSoft, Inc., Tulsa, OK, USA). In the analysis, covarion matrix was used on the normalized data to calculate principal components. Hierarchical clustering of parrot taxa was performed on the normalized morphometric features using pvclust function in R package (*R Core Team, 2015*) assuming Euclidean distance and UPGMA as agglomeration method. To estimate the uncertainty in the clustering, AU (Approximately Unbiased) *p*-value and BP (Bootstrap Probability) value were calculated for each cluster with bootstrap resampling assuming 1,000 replications. The AU *p*-value, which is computed by multiscale bootstrap resampling, is considered a better approximation to unbiased *p*-value than the standard BP value computed by normal bootstrap resampling (*Shimodaira, 2004*).

High quality parrots' vocalization files were downloaded from Avian Vocalizations Center (AVoCet, http://avocet.zoology.msu.edu) and xeno-canto database (http://www.xeno-canto.org): AV14063 (*Amazona albifrons*) recorded by Pamela C. Rasmussen, AV11523 (*Amazona agilis*) recorded by Brian K. Schmidt, XC77749 (*Amazona xantholora*) recorded by Mary Beth Stowe, XC282102 (*Amazona albifrons*) recorded by Oscar Humberto Marin-Gomez as well as XC97904 (*Amazona agilis*) and XC5942 (*Amazona xantholora*) both recorded by Richard C. Hoyer. The files together with call records obtained for the new taxon were processed and analyzed in Avisoft-SASLab 5.209 (Sound Analysis and Synthesis Laboratory) and Sound Analysis Pro 2011 (*Tchernichovski et al., 2000*), in which syllable units were identified (segmented by amplitude) and their statistic features were derived. These features were next studied by Discriminant Function Analysis (DFA) including Canonical analysis (CA) using the Statistica software (Version 1.0; StatSoft, Inc., Tulsa, OK, USA) as well as by non-parametric Kruskal–Wallis test and Dunn's test of post-hoc multiple comparisons with Benjamini–Hochberg correction for *p*-values using R package (*R Core Team, 2015*).

Silva et al. (2017), *PeerJ*, DOI 10.7717/peerj.3475

**Table 1** Morphometric data (in millimeters) of the new *Amazona* compared with other similarly red-fronted and –faced species of *Amazona*.

| Feature | New Amazona | A. a. nana | A .a. albifrons | A. a. saltuensis | A. xantholora | A. autumnalis | A. viridigenalis |
|---|---|---|---|---|---|---|---|
| Wing | 175.3 ♂ | 167.3 ♂ (Range 149.9–170.3) | 180.9 ♂ (Range 179.3–191.0) | 182.8 ♂ (Range 175.4–191.3) | 165.1 ♂ (Range 153.0–170.8 ) | | 202.4 ♂ (Range 197.1–209.1) |
| | 170.4 ♀ | 152.9 ♀ (Range 133.8–162.53) | 174.2 ♀ (Range 170.7–177.9) | 180.0 ♀ (Range 177.4–184.2) | 169.3 ♀ (Range 166.8–170.3) | 217.8[a] (Range 212.7–223.0) | 197.8 ♀ (Range 189.9–201.3) |
| Tail | 89.6 ♂ | 73.8 ♂ (Range 71.9–80.1) | 76.1 ♂ (Range 73.5–83.2) | 76.8 ♂ (Range 72.9–83.1) | 77.2 ♂ (Range 74.8–79.9) | | 104.6 ♂ (Range 89.2–117.1) |
| | 83.7 ♀ | 76.5 ♀ (Range 74.6–80.1) | 77.9 ♀ (Range 75.7–81.9) | 78.8 ♀ (Range 76.1–80.8) | 77.4 ♀ (Range 75.3–79.3) | 100.8[a] (Range 95.1–106.5) | 105.3 ♀ (Range 103.5–107.4) |
| Exposed culmen | 27.8 ♂ | 26.4 ♂ (Range 24.9–30.5) | 26.5 ♂ (Range 24.0–28.1) | 25.1 ♂ (Range 23.8–27.3) | 26.1 ♂ (Range 25.4–26.6) | | 29.9 ♂ (Range 28.5–31.7) |
| | 25.7 ♀ | 24.9 ♀ (Range 24.7–25.2) | 24.2 ♀ (Range 23.6–25.2) | 24.0 ♀ (Range 23.8–27.5) | 24.8 ♀ (Range 24.7–25.0) | 30.4[a] (Range 27.9–32.9) | 27.0 ♀ (Range 24.2–28.6) |

**Notes.**

Three individuals of each sex were measured for each taxon, except for *Amazona autumnalis* in which six birds of unknown sex were taken into account. Data were collected from living birds of known provenance and preserved skins in the collection of the Museo de las Aves de México (MAM), in Saltillo. The museum specimens are identified as: *Amazona albifrons albifrons* MAM 1076; *A. albifrons nana* MAM 2780, MAM 2217, MAM 2988, MAM 2433, MAM 1726; *A. viridigenalis* MAM 132, MAM 133, MAM 2725, MAM 1878, MAM 1548, MAM 1715, MAM 1723, MAM 1775, MAM 1377, MAM 2216, MAM 1547; *A. autumnalis autumnalis* MAM 2989, MAM 2987, MAM 1883, MAM 2448, MAM 1827, MAM 134; *A. xantholora* MAM 1948, MAM 737.

[a]Unsexed specimens.

Geographic origin of studied individuals: *A. albifrons nana*, Zoologico de Merida, from the local population; *Amazona albifrons albifrons*, Planetaro Alfa, Monterrey, from the Guerrero population; *Amazona albifrons saltuensis*, Acuario de Mazatlán (a public aquarium that also displays birds), from Sinaloa specimens; *A. xantholora*, Zoologico de Merida, from the local population; *A. autumnalis autumnalis*, Planetario Alfa, from the southern Tamaulipas population; *A. viridigenalis*, Planetario Alfa, from the southern Tamaulipas population.

**Table 2** Comparison of differences in face coloration of the new *Amazona* and other similarly red-fronted and –faced species of *Amazona* from Mexico and Mesoamerica.

| Species | Forehead | Lores | Cheeks | Crown | Black ear patch | Black scalloping contour feathers (face) |
|---|---|---|---|---|---|---|
| New *Amazona* | Red | Red | Green | Green | Absent | Present |
| *A. albifrons nana* | White | Red | Green | Bluish | Absent | Subtle |
| *A. a. albifrons* | White | Red | Green | Blue | Absent | Subtle |
| *A. a. saltuensis* | White | Red | Green | Blue | Absent | Subtle |
| *A. xantholora* | White | Yellow | Green | Bluish | Present | Present |
| *A. autumnalis autumnalis* | Red | Red | Yellow | bluish | Absent | Absent |
| *A. viridigenalis* | Red | Red | Green | Bluish | Absent | Absent |

## DNA extraction and amplification

Total genomic DNA was extracted from tail feather from the living specimens of the new *Amazona* and *A. xantholora* using Qiagen DNeasy® tissue extraction kits (Valencia, CA) and following the manufacturer's protocol. Afterwards, amplification of sex specific *CHD*-Z and *CHD*-W introns was performed for molecular sexing of the new *Amazona* individuals. The pair of 2550F and 2718R primers was used in PCR reactions according to the protocol previously described by *Fridolfsson & Ellegren (1999)*. Obtained amplicons were analyzed with the Agilent 2200 TapeStation System (Fig. S1).

Three mitochondrial genes, COI, 12S and 16S rRNA, were amplified using the previously published protocol described by *Russello & Amato (2004)*. PCR products were purified and sequenced in both directions at the sequencing service Macrogen (Rockville, MD, USA). Full complementary strands of each gene were unambiguously aligned using CodonCode Aligner (CodonCode Corporation, Dedham, MA, USA). The newly obtained sequences are available in GenBank database under accession numbers: KU605663–KU605668.

## Phylogenetic analyses

The obtained new mitochondrial sequences were aligned with all corresponding sequences of *Amazona* taxa available in GenBank, including *Pionus menstruus* as an outgroup (Table S1). Most of the sequences were obtained by *Russello & Amato (2004)* and one by *Eberhard & Wright (2016)*. Further information about geographic origin and vouchers for them is provided in their papers. The final alignment used in phylogenetic studies comprised 45 sequences with the length of 1,485 bp including three markers: 12S rRNA (390 bp), 16S rRNA (534 bp) and COI (561 bp).

For reconstructing phylogenetic trees, we applied four algorithms: Bayesian inference in MrBayes 3.2.3 (*Ronquist et al., 2012*), PhloBayes MPI 1.5 (*Lartillot et al., 2013*) and Beast 2.4.0 (*Bouckaert et al., 2014*), as well as maximum likelihood (ML) analyses in TreeFinder (*Jobb, Von Haeseler & Strimmer, 2004*) and RAxML 8.2.3 (*Stamatakis, 2014*). The best-fit partitioning schemes were selected according to PartitionFinder 1.1.1 based on BIC criterion (*Lanfear et al., 2012*)—Table S2. In TreeFinder, we also applied these partitioning scheme using models suggested by TreeFinder Propose Model module based on BIC for

Silva et al. (2017), *PeerJ*, DOI 10.7717/peerj.3475

Peer J

**Table 3** **Morphological traits of the new *Amazona* compared with other similarly red-fronted and -faced species of *Amazona*, including other species occurring in the Yucatán Peninsula (*A. albifrons nana*, *A. xantholora*, *A. autumnalis*).**

| Species | Average weight (grams) | Average length (cm) | Head coloration | Wing coloration | Tail coloration |
|---|---|---|---|---|---|
| New *Amazona* | 200 | 25 | Male: forehead and forecrown red; rear crown feathers subtle bluish tipped; periophthalmic ring red. Female: forehead red. | Underside of wings green, except tips of primaries which are bluish green. | Green, bluish tipped; three lateral tail feathers red on inner part. |
| *A. albifrons nana* | 205 (range 198.1–213.0) | 23 | Male: forehead and forecrown white, posterior border blue; periophthalmic ring and lores red. Female: white on forehead and red of periophthalmic ring greatly reduced. | Primary coverts red in male, green or red greatly reduced in most females; primaries green, dark blue towards tip; secondaries blue; under-wing coverts green. | Green, yellowish-green towards tip; base of lateral feathers red. |
| *A. a. albifrons* | 230 (range 207.4–244.4) | 26 | As *A. a. nana*, but green slightly darker. | As *A. a. nana*. | As *A. a. nana*. |
| *A. a. saltuensis* | 230 (range 211.9–233.5) | 26 | As *A. a. albifrons*, but blue crown extends to nape. | As *A. a. nana*. | As *A. a. nana*. |
| *A. xantholora* | 200 (range 197.1–238.2) | 23 | Male: forehead and forecrown white, posterior blue; lores yellow; periophthalmic ring red; ear coverts preeminently black. Female: all head colors significantly reduced, except for the crown, which is blue. | Primary and secondary flight feathers green, blue towards tip; underside of wings greenish-blue; red on shoulder present in some individuals, mainly males. | Tail green, yellowish-green towards edge; lateral tail feathers red at base. |
| *A. autumnalis autumnalis* | 350 (range 338.9–369.0) | 34 | Forehead and upper part of lores red, lower part of lores and cheeks yellow, strongly hinted with red in some individuals from Mexico; crown blue | Primary and secondary flight feathers green, becoming dark blue towards tips; first five secondaries with red wing-speculum. | Green with greenish-yellow tips; outer webs of outer tail feathers blue. |
| *A. viridigenalis* | 270 (range 266.4–299.2) | 33 | Forehead, upper lores and crown red; dark blue extends from rear part of eye and occiput to encircle cheeks, which are lighter green. Females have less red on head and some old males acquire several yellow feathers to the nape. | Outer webs of primaries violet-blue; secondaries with blue tips; first five secondaries with red wing-speculum. | Green, with green-yellow tips. |

**Notes.**
Three individuals of each sex were measured for each taxon, except for *Amazona autumnalis*, in which six birds of unknown sex were taken into account.

these partitions. Moreover, to specify appropriate substitution models across the larger space in the Bayesian MCMC analysis (*Huelsenbeck, Larget & Alfaro, 2004*), we used mixed models in MrBayes analysis.

In the MrBayes analysis, two independent runs starting from random trees were applied, each using four Markov chains. Trees were sampled every 100 generations for 10,000,000 generations. In the final analysis, we selected trees from the last 4,082,000 generations that reached the stationary phase and convergence (i.e., the standard deviation of split frequencies stabilized and was lower than the proposed threshold of 0.01). In PhyloBayes, we used CAT-GTR model with rate variation across sites modeled by five discrete rate categories of gamma distribution. The number of components, weights and profiles of the model were inferred from the data. Two independent Markov chains were run for 100,000 generations in each of these analyses. The last 85,000 trees from each chain were collected to compute posterior consensus trees after reaching convergence, when the largest discrepancy observed across all bipartitions (maxdiff) was below recommended 0.1. We set search depth to 2 in TreeFinder and applied 1,000 distinct ML searches on 1,000 randomized stepwise addition parsimony trees in RAxML. To assess significance of particular branches, non-parametric bootstrap analyses were performed on 1,000 replicates in these two programs.

Tree topologies assuming different relationships between parrots from the Greater Antilles and Central America as well as the alternative position of the newly described *Amazona* were compared according to approximately unbiased (AU), Shimodaira–Hasegawa (SH) and weighted Shimodaira-Hasegawa (wSH) tests, which were performed in Consel v0.20 (*Shimodaira & Hasegawa, 2001*) assuming 10,000,000 replicates. Site-wise log-likelihoods for the analyzed trees were calculated in TreeFinder under the best fitted substitution models.

Divergence times were estimated with Beast software. As constraints for tree calibration, we assumed the uniform prior distribution of the separation time between *Pionus menstruus* and *Amazona* dated from 5.646 to 16.553 million years ago, and the divergence time of *A. aestiva*, *A. dufresniana* and *A. pretrei* as dating from 2.877 to 10.502 million years ago, according to *Schweizer, Seehausen & Hertwig (2011)*. We tested both strict and lognormal relaxed clock models assuming the calibrated Yule model and separate substitution schemes for particular data partitions according to PartitionFinder results (Table S2). Finally, we applied the relaxed clock model for the second codon position and the strict clock model for rRNA genes as well as the first and third codon positions. The decision about the selection of clock model was made based on the inspection of the standard deviation of the relaxed clock, assuming that a value exceeding 1 indicates a significant variation among branches. The clock and substitution rates were estimated in the analyses. Posterior distributions of parameters were estimated for 100,000,000 generations with a sampling frequency of 1,000 steps. The convergence and sufficient sampling was checked using Tracer 1.6 (*Rambaut et al., 2014*). Effective sample size (ESS) for all parameters was larger than the assumed threshold 200, which indicated sufficient convergence, sampling and chain mixing. Phylogenetic trees were summarized in TreeAnnotator 2.3.1 (*Drummond et al., 2012*) with 10% burn-in of total trees using maximum clade credibility tree and

common ancestor heights. The generated tree was visualized in FigTree 1.4.2 (*Rambaut, 2012*).

The number of base differences per site (p-distance) between selected pair of sequences was calculated in MEGA6 (*Tamura et al., 2013*). The analysis involved all 1,485 positions in the alignment. The distance was expressed as percent. Standard error was estimated by bootstrap method assuming 1,000 replicates.

### Diversification rate estimation

The maximum clade credibility tree obtained from Beast and associated branching times were used for calculation diversification rate using R package LASER 2.4 (*Rabosky, 2006a*). In order to test whether diversification rates decreased with time, we calculated the $\gamma$ statistic (*Pybus & Harvey, 2000*). We also tested 11 likelihood models for diversification rates (*Rabosky, 2006b*; *Rabosky & Lovette, 2008*)—Table S3. The models were compared according to the values of the Akaike information criterion (AIC). Temporal variation in diversification rates was visualized with yuleWindow (*Nee, 2001*) within overlapping periods of 400 thousand years. The results of diversification were compared with the $\delta^{18}O$ curve (*Lisiecki & Raymo, 2005*), which is a good climate proxy. For better visualization of climate oscillations, we calculated the variance in the $\delta^{18}O$ records within the same overlapping periods.

## RESULTS

### Multivariate analyses of morphometric and vocalization features

One of the most distinctive metric features that can be used to differentiate the *Amazona* parrots from Mexico possessing red feathers in the head area are the length of the wing chord, tail and exposed culmen (Table 1). To visualize these differences, we performed a PCA analysis (Fig. 8). The first two factor coordinates explained in total 94% of variance (86% and 8%, respectively). In the PCA plot obtained, the first component is responsible for the separation of the species, whereas the second one applies to sexual dimorphism. The first component was highly correlated with all three variables: wing chord ($-0.90$), tail ($-0.96$) and exposed culmen ($-0.91$). Generally, parrots with the largest dimensions of studied characters (*A. autumnalis* and *A. viridigenalis*) are located on the left of the plot, whereas parrots characterized by smaller length values (*A. albifrons* and *A. xantholora*) are placed on the right. The new *Amazona* with the intermediate length of the wing chord, tail and exposed culmen occurs between these two extremes. The second component showed the highest correlation with exposed culmen ($-0.42$) and is responsible for the separation of sexes. Males of almost all the species are located at the lower part of the plot and are characterized by the larger length of exposed culmen than females, which are placed above them in the plot. Only the male of *A. a. saltuensis* is placed among females of other species. In fact, the difference in this parameter between *A. a. saltuensis* sexes is smallest. This taxon shows also the shortest distance between two sexes, whereas *A. viridigenalis* shows the largest. The latter species is also farthest from the parrots of Central America, which are grouped on the right site of the plot. The individual of *A. autumnalis* without assigned sex is closest to the *A. viridigenalis* male. The male of the new *Amazona* quite clearly separates

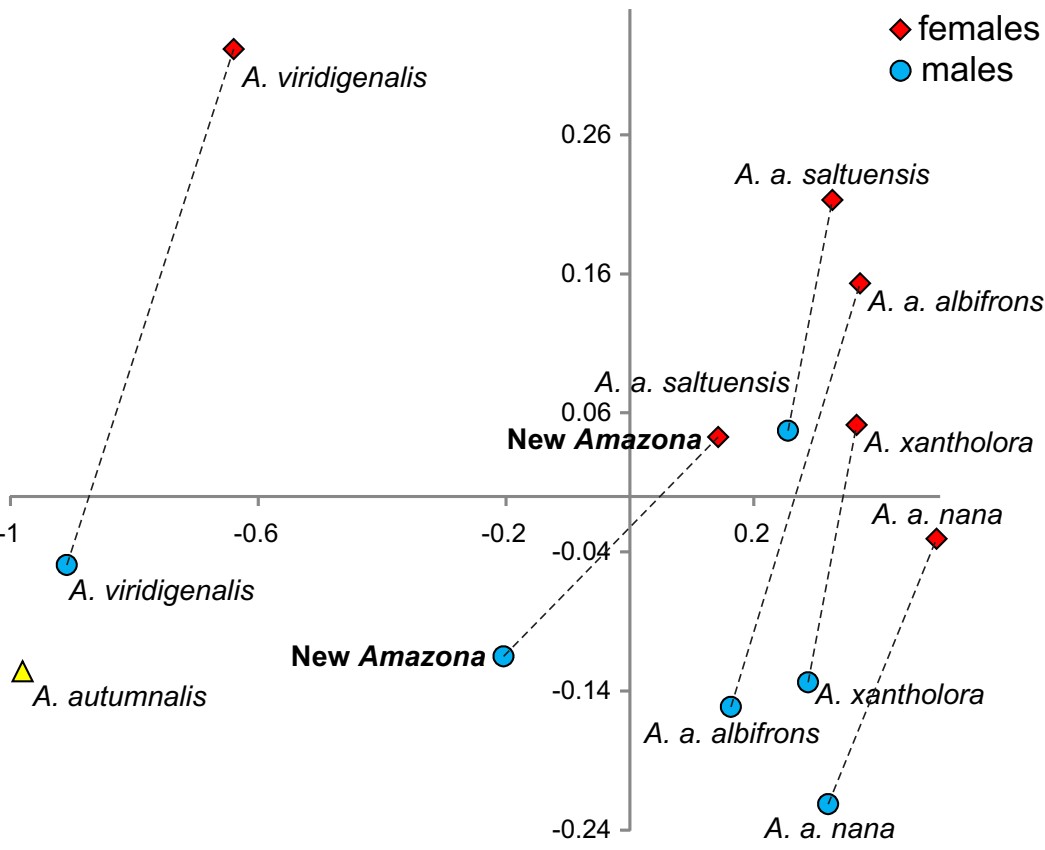

**Figure 8** The plot of the two factor coordinates from Principal Component Analysis for *Amazona* parrots displaying red in the head from Mexico and Mesoamerica separated into sexes based on three metric parameters (**length of wing chord, length of tail, culmen**). Symbols represented the same species were connected by dashed lines. The individual for *A. autumnalis* did not have assigned sex.

from males of other Mexican taxa, whereas the female of this new form is located near the *A. a. saltuensis* male and *A. xantholora* female in the plot.

We also compared the studied parrot taxa in PCA analysis (Fig. 9) using both five metric (Table 1) and six morphological features (Table 2). The first two factor coordinates explained almost 81% of variance (63% and 17%, respectively). The first component showed the highest correlation with metric features: the total length ($-0.96$), wing chord ($-0.94$), tail ($-0.94$), exposed culmen ($-0.94$) and weight ($-0.90$), as well as some morphological characters: coloring of forehead (0.84), the presence of black scalloping contour feathers (0.79) and coloring of cheeks (0.73). The second component was highly correlated with crown coloring (0.83), coloring of the lores ($-0.59$) and the presence of a black ear patch (0.59). The first component is responsible for the distinct separation of *A. autumnalis* and *A. viridigenalis* from the other Mexican parrots because of their larger weight and length of studied characters as well as the absence of black scalloping contour feathers. The Mexican parrots are differentiated by the second component into the group of *A. albifrons* subspecies and the cluster of the new *Amazona* and *A. xantholora*. The outlying position of the new *Amazona* results from its unique green coloring of crown versus blue

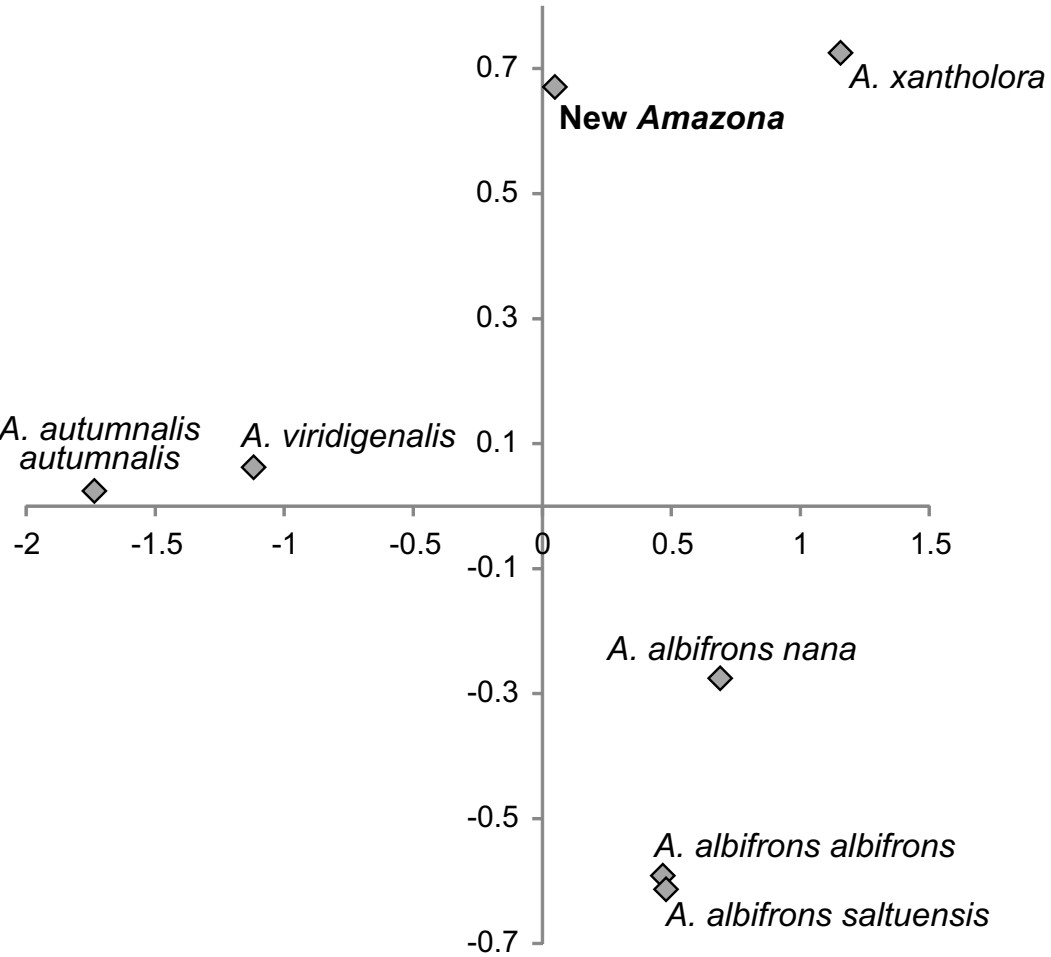

**Figure 9** The plot of the two-factor coordinates from Principal Component Analysis for *Amazona* species displaying red head feathers from Mexico and Mesoamerica based on all morphometric features.

and bluish in other parrots. In turn, *A. xantholora* separates because it has yellow lores and black ear patch as the only species of the studied species. The three subspecies of *A. albifrons* are clustered together because of white forehead and subtle black scalloping contour feathers. In agreement with these results, the hierarchical clustering based on five metric parameters clearly separates *A. autumnalis* and *A. viridigenalis* from Central America parrots (Fig. 10). At the base of the latter group, *A. xantholora* is placed and next the new *Amazona* branches off. The subspecies of *A. albifrons* create a significant cluster with *A. albifrons nana* at the base.

The proposed new taxon is characterized by a unique vocalization in comparison to other Amazon parrots inhabiting Central America (Fig. 11, Files S1 and S2). In this comparison, we also included *Amazona agilis* from the Greater Antilles because it appears the sister taxon to the Central American parrots (see section Molecular phylogenetic studies). The most distinct feature of the new *Amazona* is a relatively long duration of syllables, which is almost 5 times longer in comparison to *A. albifrons* and more than three times longer than

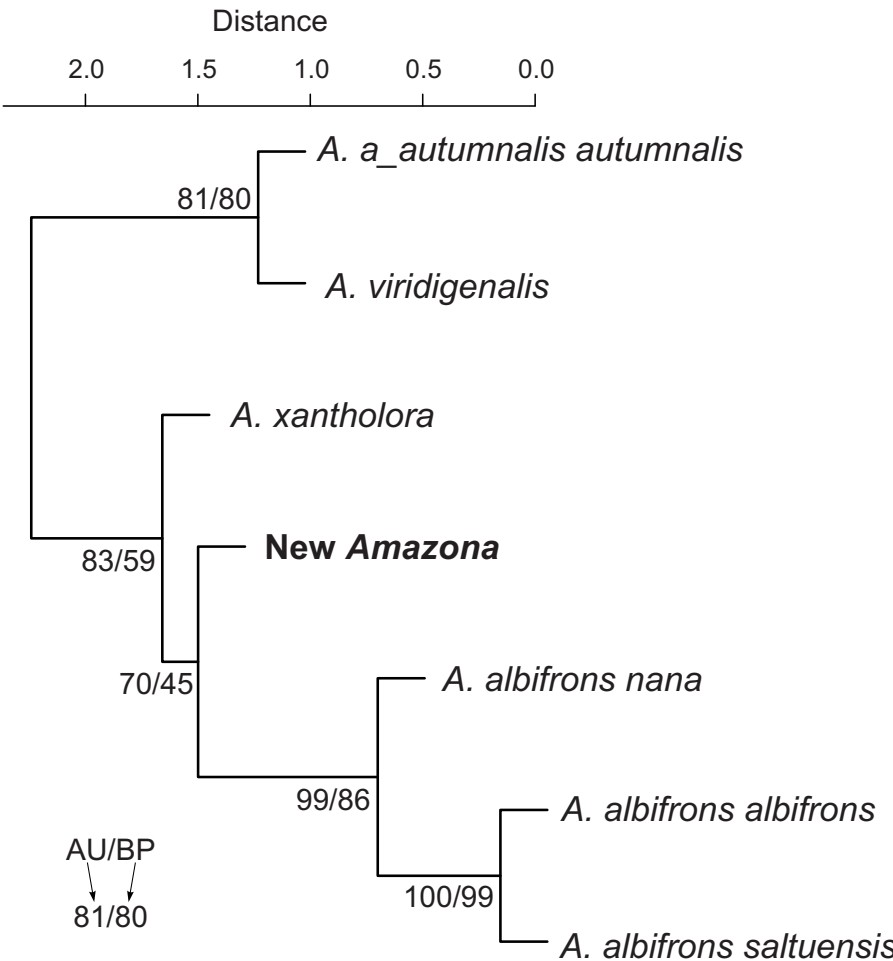

**Figure 10** **UPGMA dendrogram clustering parrot taxa according to five metric parameters (body weight and length, length of wing chord, tail and exposed culmen) and six morphological discrete characters (body weight, total length, length of wing chord, tail length, exposed culmen).** Numbers at nodes correspond to $p$-values expressed as percentages calculated using approximately unbiased test (AU) and bootstrap resampling (BP), respectively.

in *A. xantholora* (Fig. 11). For each of 12 considered vocalization features, the new taxon differs significantly ($p < 0.05$) from at least one of three other analyzed *Amazona* parrots (Fig. S2 and Table S4). Besides syllable duration, it is also significantly different from all three parrots in mean FM (frequency modulation), mean Wiener entropy (a measure of the width and uniformity of the power spectrum) and variance of mean frequency (the center of gravity of the power spectrum). In total, the new *Amazona* differs significantly in seven features from *A. albifrons*, nine from *A. xantholora* and ten from *A. agilis*.

  In agreement with that, Discriminant Function Analysis with Canonical Analysis shows the clear separation of the four parrots according to the twelve statistical features of their syllables, which indicates that they are characterized by disparate vocalizations (Fig. S3). The analysis proposes three discriminant functions (root) explaining 75.5%, 16.9% and 7.6% of variance, respectively. The first root distinctly separates *A. agilis* and the new

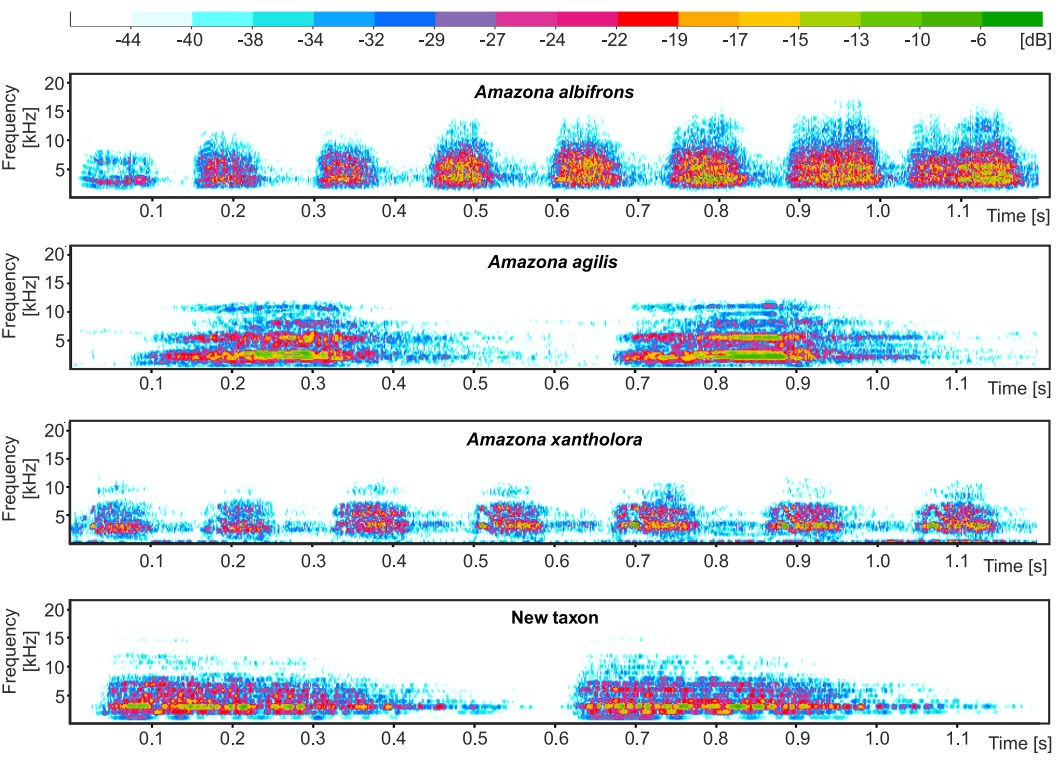

**Figure 11** Comparison of example sonogram for the new taxon with two other *Amazona* parrots from Central America and closely related *Amazona agilis* from the Greater Antilles.

*Amazona* from *A. albifrons* and *A. xantholora*. The greatest contribution (expressed by standardized function coefficients) to the first discriminant function has mean entropy (−1.210), syllable duration (−1.065), mean FM (0.969) and mean of mean frequency (0.877). Syllable duration is most correlated (−0.686) with the first root. The second discriminant function makes separate sets of syllables from *A. agilis* and the new *Amazona*, whereas the sets of *A. albifrons* and *A. xantholora* overlap partially. The second function is mostly associated with mean entropy (2.302) and mean of mean frequency (−2.227) as well as correlated with mean amplitude (−0.313) and syllable duration (−0.308). The third root separates *A. albifrons* and *A. xantholora* and is most related with variance of pitch goodness (1.184), mean amplitude (1.094) and mean pitch goodness (−1.160). The largest correlations with this function show mean $AM^2$ (−0.502) and variance of AM (−0.501).

## Molecular phylogenetic studies

Phylogenetic analyses were conducted on concatenated alignment of three genes: 12S rRNA, 16S rRNA and COI. Both Bayesian and maximum likelihood analyses showed the same quite well-resolved tree topology and relationships among the studied taxa (Fig. 12). Interestingly, none of recognized biogeographic groups (Central and South America as well as the Greater and Lesser Antilles) creates a strictly monophyletic clade that would include all members from the given region.

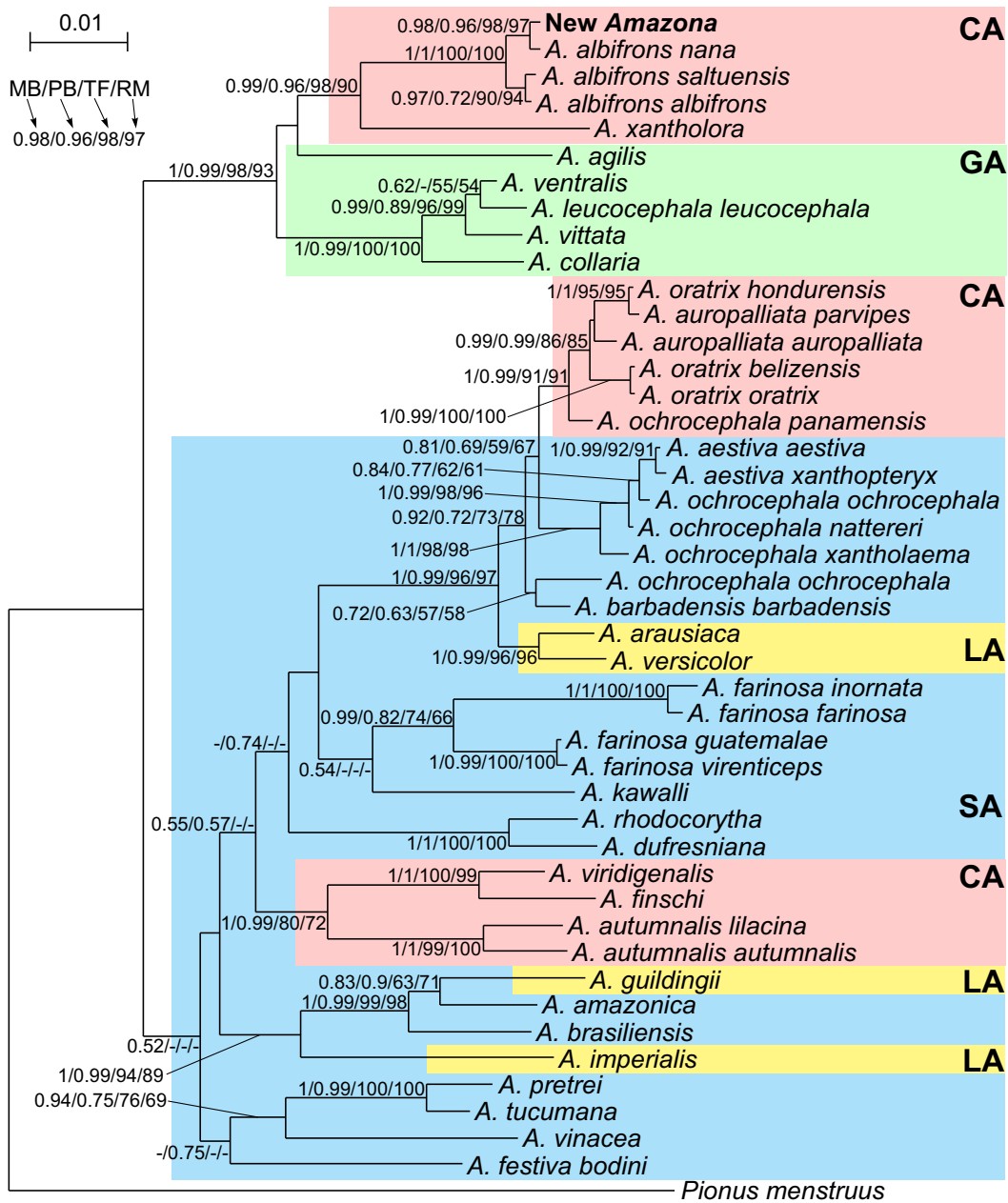

**Figure 12** **MrBayes maximum clade credibility tree for the concatenated alignment of genes for COI, 12S and 16S rRNA sequences from *Amazona* taxa and *Pionus menstruus* species (as outgroup). Numbers at nodes, in the order shown, correspond to: posterior probabilities estimated in MrBayes (MB) and PhyloBayes (PB), and bootstrap support values obtained in TreeFinder (TF) and RAxML (RM).** Values of the posterior probabilities and bootstrap percentages lower than 0.50 and 50%, respectively, were omitted or indicated by a dash "-". CA, Central America parrots; GA, Greater Antillean parrots; LA, Lesser Antillean parrots; SA, South America parrots.

The South America parrots are mixed with samples from the Lesser Antilles and Central America (Fig. 12). The Lesser Antillean parrots are clearly separated into three independent lineages. *A. guildingii* from the Lesser Antilles is significantly placed within the very distinctive group including also the South American parrots, *A. amazonica* and *A. brasiliensis*. A sister lineage to these species is *A. imperialis* from the Lesser Antilles. All four parrots form a group very well supported by all methods. The third Lesser Antillean lineage contains *A. arausiaca* and *A. versicolor*. It also obtained very high posterior probability and bootstrap values but clearly separates from the other Lesser Antillean parrots. The third lineage is very significantly related with Yellow-headed Amazon parrots from South America, namely *A. aestiva*, *A. ochrocephala* and *A. barbadensis*.

The parrots from Central America are also split into three very well supported clades (Fig. 12). The one including *A. viridigenalis*, *A. finschi* and *A. autumnalis* is placed within South America parrots. The second clade including Yellow-headed Amazon parrots is closely affiliated to their relatives from South America, namely *A. aestiva* and *A. ochrocephala* with a moderate support, whereas the third clade is very significantly grouped with the Greater Antillean parrots, i.e., *A. agilis, A. collaria, A. vittata, A. leucocephala* and *A. ventralis*.

This third clade contains parrots from Mexico and northern Central America, i.e., *A. albifrons albifrons, A. albifrons saltuensis, A. albifrons nana* as well as the newly studied *A. xantholora* and the newly described *Amazona* (Fig. 12). This clade branches off within the Greater Antillean parrots making the latter paraphyletic. The sister taxon to the Central American parrots is *A. agilis* from the Greater Antilles. The position of *A. agilis* received no support larger than 0.5 posterior probability and 50% bootstrap percentage but was indicated by all four applied methods, two Bayesian and two maximum likelihood approaches. The other Greater Antillean parrots form a clear monophyletic clade. To assess stability of phylogenetic position of *A. agilis*, we carried out tree topology tests. They showed that trees in which *A. agilis* is clustered with other Greater Antillean parrots (Fig. 13B) or placed at the base to all parrots from Central America and the Greater Antilles (Fig. 13C) were not significantly worse that the best topology (Fig. 13A).

The Mexican *Amazona* taxa are also monophyletic with *A. xantholora* placed at the base to the clade with the largest possible support including three subspecies of *A. albifrons* and the new *Amazona*. The taxa are split into two sister subclades that are well supported. One includes *A. a. albifrons* and *A. a. saltuensis*, whereas the newly described *Amazona* taxon groups with *A. albifrons nana,* with which it is sympatric. We also tested alternative topologies with different placement of the new taxon (Fig. 13). Interestingly, the tree assuming earlier divergence of the new taxon before differentiation of *A. albifrons* subspecies (Fig. 13D) was not significantly worse than the best one (Fig. 13A). However, trees with clustering the new *Amazona* to *A. xantholora* (Fig. 13E) or the basal placement of the new parrot to the rest Central America parrots (Fig. 13F) were significantly worse.

The branch leading to the new *Amazona* seems relatively short indicating a very small number of substitutions in comparison to other lineages. The number of base differences per site ($p$-distance $\pm$ standard error) expressed as percent calculated for the three markers is $0.135 \pm 0.091$ between the new *Amazona* and *A. albifrons nana*. However, it is about two

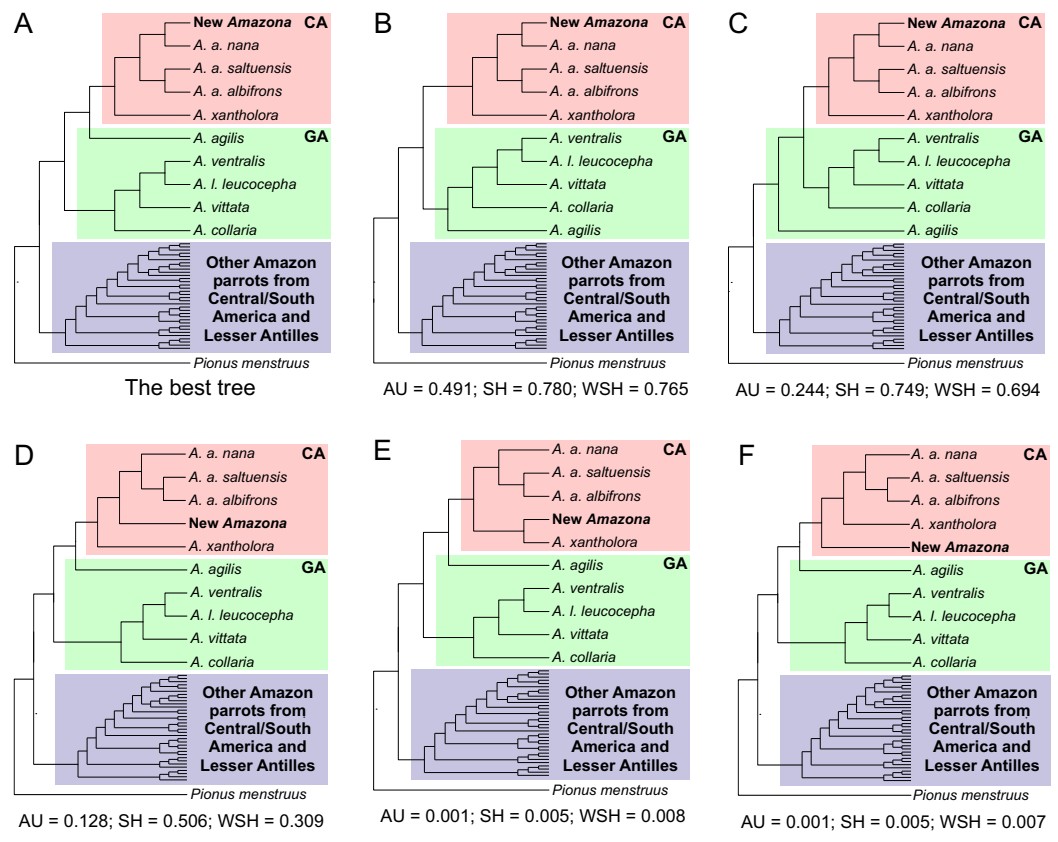

**Figure 13** **Alternative tree topologies assuming different placement of *A. agilis* (B and C) and the new *Amazona* (D, E, F) in comparison to the best found tree (A).** *P*-values of approximately unbiased (AU), Shimodaira-Hasegawa (SH) and weighted Shimodaira-Hasegawa (wSH) tests were shown. Only trees E and F are significantly worse than the best tree, whereas B, C and D cannot be rejected. SA, South America parrots; GA, Greater Antillean parrots; CA, Central America parrots.

times greater than the distance between two subspecies, *A. a. albifrons* and *A. a. saltuensis*, which is 0.067 ± 0.067. Similar conclusions can be drawn from distance calculation for individual markers but we decided to present results for the concatenated sequences because of smaller stochastic error.

The performed molecular dating enabled to estimate divergence time of important events in the evolution of Amazon parrots (Fig. 14). According to these estimations, the radiation of the present lineages of *Amazona* started about five million years ago (mya). The South American parrots begun their differentiation about 4.4 mya. The Lesser Antilles were settled from South America independently three times about 3.2, 1.5 and 1.3–0.8 mya. The South American parrots migrated also to Central America between 4.1 to 2.9 mya and also much later between 0.95 to 0.55 mya giving two separate lineages. The radiation of the third Central America clade is dated to 2.5 mya, whereas the whole group including additionally the Greater Antillean parrots started its evolution about 3.5 mya. The small number of substitution indicates quite recent divergence of the new *Amazona* from *A. albifrons nana*. Accordingly, molecular dating showed that their lineages split by average

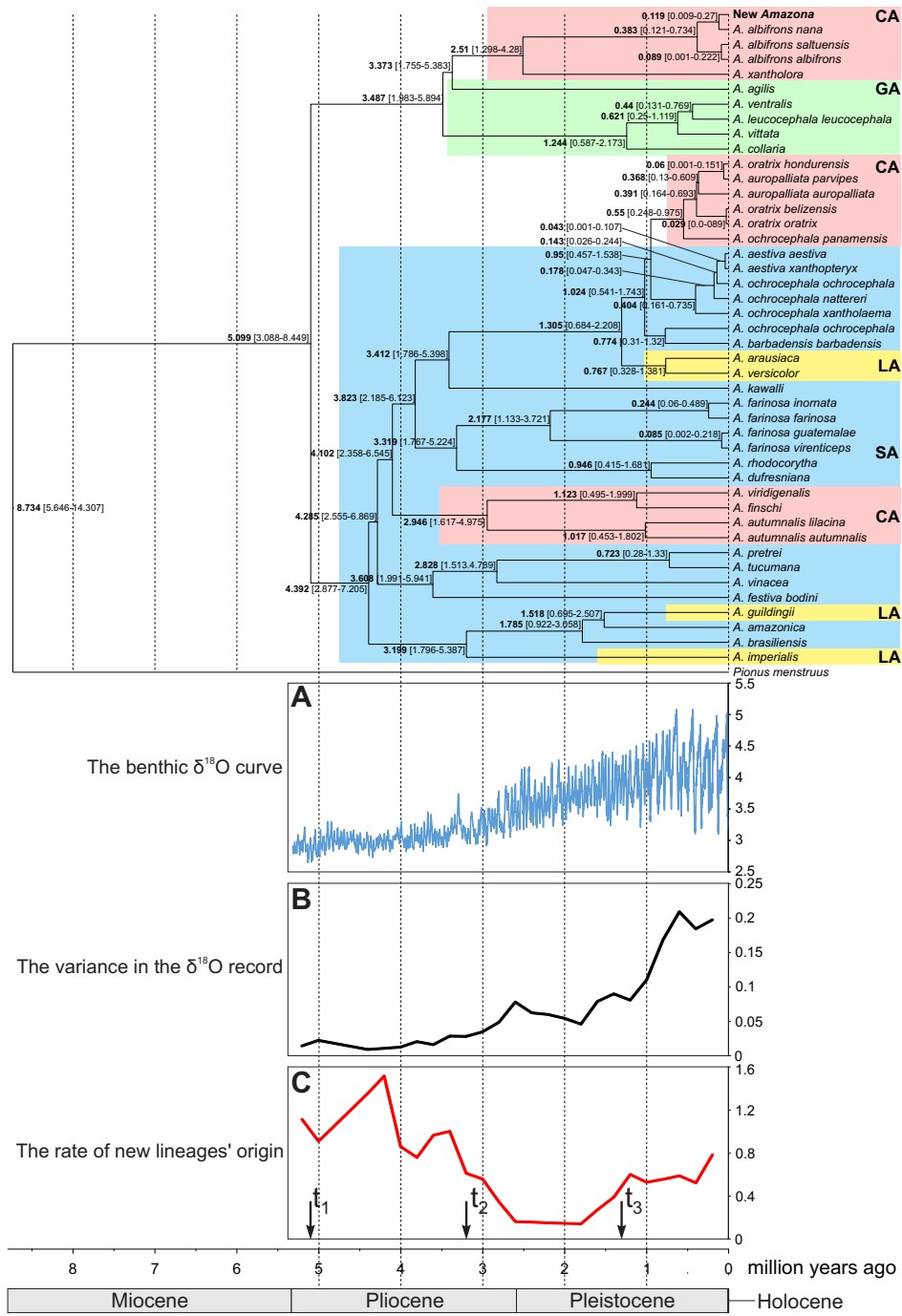

**Figure 14** **Maximum clade credibility tree obtained in Beast for the concatenated alignment of genes for COI, 12S and 16S rRNA sequences from selected *Amazona* taxa and *Pionus menstruus* species (as outgroup).** Mean (in bold) ages as well as the 95% highest posterior density distributions (in parenthesis) are shown for particular nodes. CA, Central America parrots; GA, Greater Antillean parrots; LA, Lesser Antillean parrots; SA, South America parrots. The tree was compared with benthic δ18O curve according to *Lisiecki & Raymo (2005)* (A), the variance in the δ18O records (B) and the rate of new lineages' origin (C). Arrows $t_1$, $t_2$ and $t_3$ in C indicate times in which the speciation rate shifts to a new rate according to the best-fitting yule4rate model.

119 thousand years ago (kya) with 95% credibility interval 9–270 kya (Fig. 14). The two subspecies, *A. albifrons albifrons* and *A. albifrons saltuensis* diverged slightly later about 89 kya.

## Diversification rate analyses

To assess if and when Amazon parrots (including the newly described taxon) were subjected to increase speciation rate, we performed diversification rate analyses. The calculated $\gamma$ statistic was 1.509 (*p*-value = 0.934) indicating no evidence for significant slowdown in the diversification. Among eleven tested methods, a yule4rate appeared the best-fitted (Table S3). According to this model, the first shift to a higher speciation rate (from 0.138 to 0.934) happened in $t_1 = 5.099$ mya and lasted to $t_2 = 3.199$ mya, when the rate decreased to 0.229. Since $t_3 = 1.305$ mya, the speciation rate again have increased to 0.644. The first increase is associated with radiation of the basal *Amazona* lineages (Fig. 14). The period between $t_1$ and $t_2$ corresponds to the lowest speciation of this genus. The final increase in diversification rate is related with emergence of closely related species and subspecies. Interestingly, this period corresponds to the more intensive climate fluctuations in the Pleistocene started about 2 mya (Fig. 14A). In agreement with the observation, we found significant positive correlation (Pearson correlation coefficient = 0.795 with *p*-value = 0.006) between the rate of newly diverged lineages and the variance in the climate fluctuations based on the $\delta^{18}O$ curve from 2 mya to the present (Figs. 14B and 14C).

## Description of the new taxon

As a consequence of carried out analyses, we proposed the taxonomic position of the new *Amazona*. The significant differences in morphometry, morphology, behavior and vocal features imply that the new parrot deserves species status under the typological, morphological, phenetic, as well as biological and evolutionary species concepts although genetic analyses suggest differentiation at subspecies level.

### *Amazona gomezgarzai*, sp. nov. (Figs. 2–7)

*Holotype.* Adult male, MEXICO, the Yucatán Peninsula, south of Becanchén in Tekax Municipality. The holotype is represented by the feathers of the male, which were deposited in the collection of the Laboratorio de Ornitología, Facultad de Ciencias Biologícas, Universidad Autonóma de Nuevo León, Mexico and were assigned catalog number: MGG01-*Amazona gomezgarzai*-holotipo. Article 72.5.1 of the Code of Zoological Nomenclature (henceforth CODE) permits the use of animal parts in the designation of a type specimen. Upon death of the living bird, its preserved body will be paired to the feathers for a complete body. This complies with Article 16.4.2 of the CODE, which states that where the holotype is an extant individual, a statement of the intent to deposit the individual in a collection upon its death accompanied by a statement indicating the name and location of that collection is sufficient.

*Paratype.* Adult female collected in the same locality as the holotype. Like the holotype, feathers from this specimen have been deposited in the collection and have assigned catalog number: MGG02-*Amazona gomezgarzai*-alotipo. Upon its death, it will be added to the

collection in Laboratorio de Ornitología, Facultad de Ciencias Biologícas, Universidad Autonóma de Nuevo León, Mexico.

*Etymology.* We take extreme pride in naming this parrot after Miguel Angel Gómez Garza, a Mexican veterinarian born in Monterrey (Nuevo León, Mexico) in 1960. Gómez Garza's interest in the ecology of the parrots of Mexico spans decades and culminated in the publication of a work specifically dealing with the psittacines of that country (*Gómez Garza, 2014*). During his professional lifetime, Gómez Garza has been deeply involved in rehabilitating confiscated wildlife. For the last thirty years, in his private veterinary clinic (Veterinaria del Valle) in Monterrey, he has honorably supported the wildlife protection agency of the Republic of Mexico, Procuraduría Federal de Protección al Ambiente (PROFEPA), providing medical attention to confiscated wildlife suitable for being returned to their natural habitat. As a researcher in the Facultad de Medicina Veterinaria y Zootecnia of the Universidad Autonóma de Nuevo León, he is presently working on a veterinary protocol for confiscated psittacines intended for reintroduction to the wild. He brought the existence of this unique member of the genus *Amazona* to our attention and to him science and we owe a debt of gratitude. We suggest the common name in English: blue-winged Amazon and in Spanish: Loro de alas azules.

*Diagnosis.* The studied specimens show all of the characteristics of the genus as described by *Lesson (1831)*: "Rugged beak, very hooked, thick, banded edge or forming a flattened depression, narrow, which follows the curvature of the beak, swollen sides, the scalloped edges; the fringed lower mandible forward; nostrils rounded, very open in the wax and with protruding flange; wings extending until one third of the tail; the tarses very short, reticulate, robust." Their behavior, including display, is consistent with that of the genus and is closer to *A. viridigenalis* than *A. xantholora* or *A. albifrons,* the birds being active and very vocal (T Silva, pers. obs., 2015; MA Gómez Garza, *in litt.*, 2015).

A very distinctive feature of the new taxon is its call, which is loud, sharp, short, repetitive and monotonous; one particular vocalization is more reminiscent of an *Accipiter* than of any parrot we know (Files S1 and S2). In flight, the call is a loud, short, sharp and repetitive *yak-yak-yak* that is never repeated in pairs like in *A. xantholora.* While perched, the call is mellow and prolonged, sharper and more melodious than that of *Amazona albifrons.* Perched birds always respond to the call of another flock member, insuring that the parrots always maintain contact with one another.

In general appearance, the new *Amazona* demonstrates a similarity to *A. vittata* of Puerto Rico and to a lesser extent to *A. tucumana* from Bolivia and Argentina and *Amazona pretrei* from Brazil and Argentina. *A. vittata* can be distinguished from the female of the new *Amazona* by the paler colored bill, larger and purer white orbital ring and more prominent grey bordering to the feathers. Male of the new *Amazona* can be separated from *A. vittata* in addition by the presence of rose-red feathers around the orbital ring.

*A. pretrei* exhibits dimorphism like the new *Amazona* but the male of *A. pretrei* displays significantly redder feathering on the head and considerable red on the bend of the wing and carpal edge; the red tone in *A. pretrei* is also richer. Indeed, both sexes of *A. pretrei* possesses more extensive red feathering in the head, the color extending to forecrown and covering a broader area around the orbital region; red feathers tend to appear scattered on the head;

the bend of wing and carpal area are covered in red as previously mentioned; the grey bordering to the feathers is more prominent; and the bill is smaller, more proportionate and tends to have an orangish hint, a color which intensifies with the breeding cycle.

The new *Amazona* is more phlegmatic in its behavior compared to the highly excitable and vocal *A. pretrei*. In turn, *A. tucumana* differs by having a reddish triangle on the head, extending from forehead to crown (a feature seen only in male of the new *Amazona*); there is an absence of dimorphic head coloration; the grey bordering to the feathers is very prominent; the head is more proportionate and the bill is whitish. In behavior, there are some affinities between *A. tucumana* and the new *Amazona*.

Of the Mexican species, the new *Amazona* can easily be separated from *A. xantholora* by the absence of yellow, white and blue from the head, from the green ear coverts and by the absence of the prominent barring to the body feathers. The new taxon can be differentiated from *A. albifrons* by the absence of white and blue from the head and by the green alula in both sexes, as well as a larger size when compared to the sympatric *A. albifrons nana*.

The new *Amazona* resembles *A. albifrons albifrons* in size. Although its general color scheme is closer to *A. viridigenalis* from northern Mexico (Table 2), the size difference is significant and diagnostic as pointed out in Tables 1 and 3. Moreover, the new *Amazona* has less red on the head and possess neither the distinctive yellowish nape feathers that appear in elderly male *A. viridigenalis* nor the red wing speculum found in *A. viridigenalis*.

### Description

*Male (holotype).* Total length 26.6 cm; wing (chord) 175.3 mm; exposed culmen 27.8 mm; tail 89.6 mm. The sex has been determined with molecular methods (Fig. S1). Upperparts, including nape, auriculars, dorsum, tertials, wing-coverts, rump and upper tail coverts parrot green, the feathers of the head, nape, neck and mantle subtlety bordered black; forehead, forecrown and feathers surrounding naked periophthalmic ring rose-red. Rear crown feathers subtlety bordered in blue. Underside, chin, throat, breast and belly parrot green, the feathers from chin to vent subtlety bordered in blue. Thigh feathers also washed in blue. Primaries (numbered descendently) dark blue with flight feathers numbers 10 and 9 green on outer webs near base. Secondaries blue with green margin on outer webs. Primary coverts blue, except along shaft, which is green. Upperside of tail: two central feathers green; other feathers blue on outer web, red on inner web, then yellowish green towards tip; all tail feathers are bordered in blue. Underside of tail: two central feathers green, reddish hinted near feather shaft; other tail feathers rose-red on inner web, yellowish at base and green towards tip; outer webs greyish-blue. Bill yellow, whitish at tip of upper mandible. Tongue flesh grey, exposed nares naked and grey colored, periophthalmic ring naked and greyish-white colored, iris pale mustard yellow, feet greyish-flesh colored and nails grey, darkest at tip.

*Female (paratype).* Total length 24.7 cm; wing (chord) 170.4 mm; exposed culmen 25.7 mm; tail 83.7 mm. The sex has been determined with molecular methods (Fig. S1). Like male but rose-red confined to forehead.

## Additional characteristics

*Distribution.* The new *Amazona* is endemic to the Yucatán Peninsula in southern Mexico. To date, its presence is confined to an area roughly 100 km$^2$ that is centered south of Becanchén in Tekax Municipality, Yucatán. No part of the range is presently protected in any form.

*Habitat.* The new *Amazona* is found in tropical caducifolius and subcaducifolius forest. It is also found in disturbed patches of native vegetation and in small, cultivated fields with scattered trees. It is found below 300 m above sea level.

*Natural history.* Miguel A. Gómez Garza first sighted this parrot in the field in trees of the *Leucaena* genus at heights of approximately 6 m in the beginning of 2014 during a visit to the south of Becanchén, in the municipality of Tekax. The parrots occurred in small flocks of three to five individuals and fed on the tender pods produced by this tree. During a follow up visit in August 2014, Gómez Garza also sighted pairs with their fledged young. This field work confirmed the rarity of the species and that it was far less common than the other two species found in the same area, *Amazona albifrons nana* and *Amazona xantholora*.

In normal parrot fashion, the new *Amazona* is diurnal, beginning the day at sunrise. It is generally secretive when resting, using its plumage as camouflage. In contrast, it is vocal and noisy in flight. The flight is moderately fast with the mechanism that is typical of the genus *Amazona* with wing-beats never exceeding the horizontal axis.

The new *Amazona* is found in small flocks of less than 12 individuals, which were studied in the field. Pairs and their progeny have a tendency to remain together and are discernible in groups. Like all members of the genus *Amazona*, this parrot is herbivore. Its diet consists of seeds, fruits, flowers and leaves obtained in the tree canopy. It also consumes tender shoots of native trees and the pods of leguminous trees including uaxim (*Leucaena glauca*), bukut (*Cassia grandis*) and katsín (*Acasia gaumeri*).

Very little is known about this parrot's biology. There is no conservation program currently in effect to preserve this parrot but its long-term existence impinges on the local communities and making them aware of this parrot's value as a result of its uniqueness, its potential as a bird watching attraction and the fact that it is present only locally. Its small range and rarity should make its conservation a priority.

## DISCUSSION

### Taxonomic position of the newly described *Amazona*

In this study, we proposed the new taxon of *Amazona* at the species level, *Amazona gomezgarzai* sp. nov. The species level is supported by morphometrical and behavioral data, whereas mitochondrial genetic analyses imply the subspecies level. Below we discuss the pros and cons of these two taxonomic concepts in an objective way.

Multivariate analysis incorporating both metric and morphological features clearly separated the new *Amazona* from the other Mexican parrots, which in turn differed distinctly from *A. autumnalis* and *A. viridigenalis* (Figs. 8–10 ). The newly described taxon showed the closest morphometric similarity to *A. xantholora*. However, it clearly

separates in vocalization features from two other Central American parrots (*A. albifrons* and *A. xantholora*) and their Greater Antillean relative *A. agilis* (Fig. 11 and Fig. S3).

Based on the phylogenetic analysis performed, this new taxon is undoubtedly grouped within the clade of Mexican congeners, namely *Amazona albifrons albifrons, A. a. nana, A. a. saltuensis* and *A. xantholora* (Fig. 14). Although the new *Amazona* shows some morphological similarity to *A. vittata*, these two taxa are clearly separated into two clades in the phylogeny. The closest relative of the new *Amazona* is *A. a. nana*, also from the Yucatán Peninsula. The other two subspecies of *Amazona albifrons* (*A. a. albifrons* and *A. a. saltuensis*) included in the same clade are distributed along the Pacific slope of Mexico (*Gómez Garza, 2014*) and surely share the same ancestors as the forms from the Yucatán Peninsula. The common origin of these taxa, along with the newly described form, is concordant in several common features, such as sexual dimorphism and similar plumage coloration with the presence of red on the head (Fig. 3).

Nonetheless, the new *Amazona* is clearly different from *A. albifrons*. Although the three Mexican parrots found in the Yucatán Peninsula (*A. albifrons nana, A. xantholora* and the new *Amazona*) share the same habitat and come into frequent physical contact, they live commensally and show substantial differences in their morphology, plumage, call and behavior (Tables 1–3, Figs. 3 and 11, Fig. S3). The features used here to discriminate the proposed taxon are of the same type as those utilized in elevation of other *Amazona* species. The characters described allow the species to live sympatrically without hybridizing (*Pettingill, 1970*). This suggests that these three forms are separate species. The differentiation in characters involved in mate choice, such as song, plumage, and behavior play a central role in avian speciation (*Edwards et al., 2005*). The role of song is particularly interesting because multiple factors influence vocal evolution and this feature is subjected to rapid change through learning and behavioral evolution.

Studies of geographic variation in the vocalizations of the crimson rosella (*Platycercus elegans*) parrot species complex showed that vocal variation, in a species with vocal learning, can coincide with areas of restricted gene flow across geographically continuous populations. These results suggest that vocalization can be associated with reduced gene flow between populations, and therefore may promote speciation, even in the absence of other barriers (*Ribot et al., 2012*). On the other hand, several local dialects were documented for *Amazona auropalliata* with no significant relationship with genetic variations (*Wright & Wilkinson, 2001*) indicating a high degree of gene flow and individual dispersal across the dialect boundaries. Experimentally simulated dispersals with *Amazona auropalliata* individuals moving within and across dialect regions showed that both vocal learning (in the case of juveniles) and limited dispersal (in the case of adults) are responsible for the dialect maintenance (*Salinas-Melgoza & Wright, 2012*). Although recent studies on contact calls of Neotropical parrots from the tribe Arini (related to Androglossini) showed evolutionary rates similar (but not accelerated) to those of morphological traits, the calls contained significant levels of phylogenetic signal and evolution of some acoustic parameters correlated with evolution of body mass and bill length (*Medina-Garcia, Araya-Salas & Wright, 2015*). The coordinated evolution of these features can facilitate speciation of parrots.

On the other hand, it could be possible that the studied individuals of the new taxon are hybrids or aberrant forms of *Amazona albifrons* and the observed morphometric differences result from intraspecific variation in *A. albifrons*. However, the length of wings and tail of the newly described parrots are out of the range of these characters in all three *Amazona albifrons* subspecies. The red forehead, green crown and distinct black scalloping contour feathers were not observed in *A. albifrons* too. Field studies carried out by Miguel A. Gomez Garza and others during the past 30 years have revealed no individuals of *A. albifrons* showing such mixed characters. Similarly, local informants and the staff at PROFEPA (Procuraduría Federal de Protección al Ambiente) and CIVIS (Center for the Conservation and Research of Wildlife), which is managed by the government in the same municipality of Tekax, have never seen such potential hybrids or *A. albifrons* with the atypical features among the hundreds of all parrots confiscated in the range area each year, either. Such forms were not observed also among the hundreds of all parrot specimens imported through the US quarantine system from 1973 to 2008, when a ban was introduced on export of parrots.

The Kawall's Amazon (*Amazona kawalli*) was also initially considered an aberrant form of Mealy Parrot (*Amazona farinosa*) before it was recognized as the new species (*Martuscelli & Yamashita, 1997*). Nevertheless, more extensive studies including larger number of *A. albifrons* specimens are necessary to verify its variation because aberrant forms are not unusual in parrots.

The distinct morphological and behavioral features seem incongruent with molecular phylogenetic results, in which the new *Amazona* and *A. albifrons nana* are clustered together leaving outside the two *A. albifrons* subspecies. It would suggest that the new taxon should be a subspecies within *A. albifrons*. However, the alternative placement of the new *Amazona* at the base to the monophyletic *A. albifrons* clade is not significantly worse than the best tree (Fig. 13D). It suggests that the new taxon could have emerged before differentiation of *A. albifrons* to subspecies and has reached a species level. Interestingly, such alternative topology was obtained for hierarchical clustering of parrots based on all morphometric characters (Fig. 10). Moreover, the molecular distance between the new *Amazona* and *A. albifrons nana* measured by the number of base differences per site (0.135) is even about two times greater than the distance (0.067) between two *A. albifrons* subspecies, *A. a. albifrons* and *A. a. saltuensis*.

The acceptance of the new *Amazona* as a species would imply that the *A. albifrons* taxon would be paraphyletic. In consequence, *A. albifrons nana* could be also admitted a species status. However, it is not sufficiently different in morphology and morphometry from other subspecies of *A. albifrons* to be elevated to the new species. It should be noted that the paraphyly of *Amazona* taxa is not an exceptional case because the same situation concerns Central American *A. oratrix* and *A. auropalliata*, whose sequences are mixed and do not form one-species monophyletic clades (Fig. 12). Similarly, *A. ochrocephala* is also paraphyletic whose representatives group with *A. aestiva*, *A. barbadensis* and the clade *A. oratrix* - *A. auropalliata*. It cannot be excluded that some specimens (e.g., *A. ochrocephala*) were misidentified and the taxonomy of the genus *Amazona* should be substantially revised.

The resulted paraphyly of *A. albifrons* caused by the new *Amazona* does not have to be an extraordinary case, either. The comprehensive surveys and meta-analyses of mitochondrial gene phylogenies pointed out that such paraphyletic or polyphyletic species constitute a substantial fraction (19–23%) of thousands animal taxa studied, including four parrot species from the *Cacatuidae* family (*Funk & Omland, 2003*; *Ross, 2014*). The major natural reasons of species-level paraphyly and polyphyly can be introgression and incomplete lineage sorting following recent speciation. However, following Haldane's rule (*Haldane, 1922*), the introgression of maternally inherited mtDNA is restricted between heterogametic avian species because female hybrids are characterized by a reduced viability (*Brumfield et al., 2001*; *Carling & Brumfield, 2008*; *Rheindt & Edwards, 2011*; *Saetre et al., 2001*; *Saetre et al., 2003*; *Tegelstrom & Gelter, 1990*; *Turelli & Orr, 1995*). Mitochondrial genes are also less prone to the incomplete sorting than nuclear loci because they are present in a haploid genome and maternally inherited (*Hudson & Turelli, 2003*). However, this effect can influence mtDNA in rapidly radiating taxa, in which on-going speciation occurs before genetic sorting (*Funk & Omland, 2003*).

Assuming that the current phylogeny reflects real relationships between Amazon parrots, we can accept that the paraphyletic species, including the new *Amazona*, have emerged quite recently within other species from their isolated subspecies. In the case of the new *Amazona*, its lineage diverged most probably about 120,000 years ago within *A. albifrons* (Fig. 14). During this time, the taxon differentiated sufficiently to be clearly recognizable by many morphometric and behavioral features. In agreement with that, the genetic distance between the new *Amazona* and *A. a. nana* is two times larger than that between their closest relatives *A. a. albifrons* and *A. a. saltuensis*.

The taxon described here, morphologically and behaviorally different from other members of the genus found in Mexico, is not an exception regarding the small genetic distance. There are many examples of birds with minor genetic differences that are treated as valid species, e.g., *Apus apus*/*A. pallidus* (*Päckert et al., 2012*), *Clanga clanga*/*C. pomarina* (*Helbig et al., 2005*; *Lerner et al., 2017*) and *Falco rusticolus*/*F. biarmicus*/ *F. cherrug* (*Nittinger et al., 2007*). Recent estimates of avian diversity suggest that the current taxonomy of birds underestimates their species number by at least a factor of two (*Barrowclough et al., 2016*). Subsequent studies of the new *Amazona* should be carried out to deliver additional information about this interesting parrot.

## Implication on general phylogeny and migration of Amazon parrots

Our results have also interesting implications for phylogeography of the whole genus *Amazona* and colonization of Central America as well as Lesser and Greater Antilles. The obtained results indicate that Central America was settled three times independently at different times from distinct ancestral lineages. Two times their ancestors were South American parrots and the immigrations happened 4–3 mya and 1–0.5 mya. It is in good agreement with the standard assumption on the formation of the Panama Isthmus, whose final closure is proposed to have occurred just 4–3 mya (see *Montes et al., 2015*) for the much earlier dating, which also supports our estimations). The third case is more controversial because the clade does not cluster directly with any South American parrots but with those

from Greater Antilles. The observed proximity of the Mexican *Amazona albifrons* clade with the Greater Antillean clade composed of *Amazona collaria, A. vittata, A. leucocephala, A. ventralis* and *A. agilis* suggests the continental origin of the island parrots (*Bond, 1963*; *Lack, 1976*; *Lantermann, 1997*; *Ottens-Wainright et al., 2004*; *Russello & Amato, 2004*; *Snyder, Wiley & Kepler, 1987*; *Wiley, 1991*). Two colonization events of the Greater Antilles from Central America, i.e., Yucatan Peninsula and Honduran-Nicaraguan Bulge were proposed (*Bond, 1963*; *Lack, 1976*; *Lantermann, 1997*; *Snyder, Wiley & Kepler, 1987*; *Wiley, 1991*). It was hypothesized that one invasion could have occurred through Jamaica (by lineage of *A. agilis*) and the second through Cuba (by *A. leucocephala* from which other Jamaica parrot *A. collaria* would derive) (*Lack, 1976*; *Lantermann, 1997*; *Snyder, Wiley & Kepler, 1987*; *Wiley, 1991*). *Ottens-Wainright et al. (2004)* proposed also two colonization events but both directed to Jamaica.

Our phylogenies including the largest number of *Amazona* representatives from Central America do not split the Greater Antillean parrot clade into two groups as would be expected in the case of the two-colonization scenario. Just the opposite, they show that the Central America clade is nested within the Greater Antillean parrot group. Such branching order results from the basal position of the quite diverged lineage of *A. agilis* to the Central American parrots. In the LogDet model-based tree by *Ottens-Wainright et al. (2004)*, the consensus of 12 equally most parsimonious trees by *Russello & Amato (2004)*, and Bayesian Beast tree by *Schweizer et al. (2014)*, the Central America clade was also placed within the Greater Antillean parrot group but in these cases *A. agilis* was basal to both Central America and Greater Antilles clades. These two alternative topologies are not statistically different but the first one is favored (Figs. 13A and 13C). The topology assuming the separation and monophyly of the Central American and the Greater Antillean parrots was also not rejected by the applied tests (Fig. 13B). However, when taking into account that the first topology (Fig. 13A) was inferred by all four applied methods and the Greater Antillean parrot clade, including *A. agilis,* shows a greater genetic variation and older divergence time than the Central America clade, it is possible that a migration happened from the Greater Antilles to Mexican territory. In this scenario, the Greater Antillean parrots would be derived from species inhabiting northern South America, whose lineages became extinct and therefore are not present in inferring phylogenies. According to our molecular dating, the colonization of the Central America could happen between 3.4 to 2.5 mya (Fig. 14). These event is in agreement with dating of decrease in sea level, which started to systematically fall since 3 mya and in the period 3.4 to 2.5 mya descended even 50 m below the present level (*Hansen et al., 2013*), which could have facilitated the migrations.

In the case of the Lesser Antillean parrots the situation seems clearer. The presence of three separated clades placed within South American parrots suggests independent migrations from the mainland to the islands as proposed by *Bond (1963)*. Our estimations indicate that it could have happened about 3.2, 1.5 and 1.3–0.8 mya, which well correspond with the decrease in sea level initially by 25 m and after 2.5 mya by more than 50 m with relation to the Pleistocene glaciations (*Hansen et al., 2013*). However, we cannot exclude the opposite direction of migrations, from islands (the Lesser Antilles) to the mainland (the northern coast of Venezuela) as it was recently proposed for the origin of Yellow-headed

Amazon parrots (*Urantowka, Mackiewicz & Strzala, 2014*). Nevertheless, the obtained results show a complex history for parrots within the Caribbean region (*Russello & Amato, 2004*) related probably with the refugial and insular character of its habitats. Additional studies are required to solve in detail the migration routes.

The Amazon parrots have been subjected to evolutionary expansion since the last 5 mya. Their earliest diversification may be associated with adaptive radiation which has beentriggered by the arrival of Arini parrots in South America from Africa (*Schweizer et al., 2014*). Other important factors could be drainage evolution in Amazonia and Pleistocene climatic oscillations (Fig. 14) causing alterations and partitioning of habitats, sea level changes influencing colonization of islands (and again mainland) as well as recurrent elevational migrations (*Ribas et al., 2012*; *Rull, 2011*; *Schweizer et al., 2014*). These processes could cause the differentiation of populations into new lineages. One of such recently evolving lineage could represent the newly described Amazon parrot. *Schweizer et al. (2014)* studying the diversity of Neotropical parrots (including members of Arini and Androglossini clades) found no evidence of the slowdown in their speciation rate and discovered two young, unexpectedly species-rich clades represented by *Pyrrhura* and *Aratinga*. Although these two clades originated in the late Miocene/Pliocene, speciation within each clade took place mainly during the Pleistocene. The same can be observed in the case of at least some *Amazona* lineages (Fig. 14).

## CONSERVATION ASPECTS

If the newly described *Amazona* represents the species status must be regarded as critically endangered (CR) based on IUCN (International Union for the Conservation of Nature) Red List of Species criteria, as all new species described in recent decades. Its habitat has been significantly altered. This parrot is confined to a small area and no parts of its range are currently protected. Because of this precarious status, the Mexican wildlife authorities are urged to regard it as "Especie en Peligro de Extinción" (Endangered species), in following with established guidelines (Norma Oficial Mexicana NOM-059-SEMARNAT-2010). This *Amazona* does not undergo displacement, making them confined to a small area of lowland native forest and interspersed altered plots containing native vegetation. Through the publication of this description, we are alerting government authorities, conservationists and local inhabitants that implementing conservation measures is imperative to provide refuge for a broad array of species found within the range of *Amazona gomezgarzai*, including this unique new member of the genus *Amazona*. Because of this precarious status, the Mexican government would not allow the collection of voucher specimens. Instead, the authorities permitted that two individuals maintained locally as pets be transported for safe keeping under the care of Dr Miguel Angel Gómez Garza.

## ACKNOWLEDGEMENTS

We would like to thank the Mexican authorities at Procuraduría Federal de Protección al Ambiente (PROFEPA) but in particular the former head of Natural Resources, Alejandro del Mazo Maza, as well as Ana Romo, Joel González and Saúl Colín for their cooperation

in procuring specimens. Molecular sexing and mtDNA sequence studies were carried out by Ricardo Canales of the Laboratorio de Biología de la Conservación y Desarrollo Sustentable, Facultad de Ciencies Biologicas, Universidad Autonóma de Nuevo León and to him we are most grateful. We also express our appreciation to José I. González Rojas, chief of the Department of Ornithology of the same institution, who provided access for the study and measurement of comparative material. Aldegundo Garza de León of the Museo de las Aves de México in Saltillo, gave us unfettered access to the collection. Juan García Venegas did the illustrations. Miguel A. Pérez Hassaf, Eduardo Serio, Ricardo Cantú López, Edgar Villarreal, Daniel Garza Tobón, Carlos Leal, Jorge Verduzco, and Roberto Chavarría provided comments and suggestions to improve this paper. Finally, the late Ramon Noegel and Helmut Sick instilled a passion for these parrots that to this day pervades in our soul. We are very grateful to Norbert Bahr and other reviewers for their valuable comments and insightful remarks that significantly improved the paper.

### Funding

Studies were supported by the National Science Centre Poland (Narodowe Centrum Nauki, Polska) grant no. 2015/17/B/NZ8/02402. Publication costs were supported by the Wroclaw Center of Biotechnology program "The Leading National Research Center (KNOW) for years 2014-2018". The funders had no role in study design, data collection and analysis, decision to publish, or preparation of the manuscript.

### Grant Disclosures

The following grant information was disclosed by the authors:
National Science Centre Poland (Narodowe Centrum Nauki, Polska): 2015/17/B/NZ8/02402.
Wroclaw Center of Biotechnology program.

### Competing Interests

The authors declare there are no competing interests.

### Author Contributions

- Tony Silva, Antonio Guzmán and Adam D. Urantówka conceived and designed the experiments, performed the experiments, analyzed the data, contributed reagents/materials/analysis tools, wrote the paper, prepared figures and/or tables, reviewed drafts of the paper.
- Paweł Mackiewicz conceived and designed the experiments, performed the experiments, analyzed the data, contributed reagents/materials/analysis tools, wrote the paper, prepared figures and/or tables, reviewed drafts of the paper, managed the submission, responded to editor and reviewers as well as prepared the final manuscript.

## Animal Ethics

The following information was supplied relating to ethical approvals (i.e., approving body and any reference numbers):

Procuraduría Federal de Protección al Ambiente (PROFEPA), the national wildlife protection agency; Laboratorio de Ornitología, Facultad de Ciencias Biológicas, Universidad Autonóma de Nuevo León, Mexico.

## DNA Deposition

The following information was supplied regarding the deposition of DNA sequences:

GenBank accession numbers: KU605663–KU605668.

## Data Availability

The raw data has been supplied as a Supplementary File.

## New Species Registration

The following information was supplied regarding the registration of a newly described species:

Amazona gomezgarzai LSID: urn:lsid:zoobank.org:act:C4AA8659-8077-4195-9E11-D2EB3635397C.

Publication LSID: urn:lsid:zoobank.org:pub:60A61384-B6B1-4DEF-8B2A-D0AA74E60339.

## Supplemental Information

Supplemental information for this article can be found online at http://dx.doi.org/10.7717/peerj.3475#supplemental-information.

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
