# Peer review of "A new parrot taxon from the Yucatán Peninsula, Mexico—its position within genus Amazona based on morphology and molecular phylogeny"

_PeerJ, doi:10.7717/peerj.3475_

## Round 0.1 · original submission · Major Revisions

Dear authors,

As you can see our reviewers would like to see some improvement of your MS. I hope that you can revise the ms accordingly. We will send the revision to the original reviewers.

Kind regards,

Michael Wink
Academic editor

Reviewer 1 ·

Basic reporting

no comment

Experimental design

no comment

Validity of the findings

no comment

Additional comments

I commend the authors for their extensive data set. In addition, the manuscript is clearly written in professional, unambiguous language. It is visible, that the manuscript underwent several stages of review, so the manuscript is well developed with a huge amount of methods and data given. Sometimes, it is not clear if the focus is still on the new taxon or on the phylogeny of the Central American Amazona genus.
Some minor changes should be made before acceptance.

line 101 please write “auropalliata” with double “l”

lines 656-659: “Similarly, local informants and the staff at PROFEPA have never seen such hybrid among the hundreds of parrots confiscated in the range area each year, either. The same can be said about the hundreds of specimens imported through the US quarantine system from 1973 to 2008, when a ban was introduced on export of parrots.” – When you write about “hundreds of parrots/specimen” confiscated or imported through the US quarantine system, do you refer to the new taxon or to all confiscated parrots in the range area of 100 km² or to all parrots except the new taxon? Please make it clear. It seems that the new taxon could have already been exported in many individuals into the US.

Fig. 11: what is the difference between the five metric characters (body weight and length, length of wing chord, tail and exposed culmen) and the six morphological characters (body weight, total length, length of wing chord, tail length, exposed culmen)? They seem identical to me as five characters
six morphological characters (ll. 192-194): “coloration of forehead, lores, cheeks and crown, the presence of black ear patch and black scalloping on contour feathers on the face.”

Fig. 14: Phylogenetic tree "A. autumnalis lilacine" should be "lilacina". A. ochrocephala ochrocephala appears in 2 clades, first in A. aestiva and second in A. barbadensis clade. Please explain.

Reviewer 2 ·

Basic reporting

I commend the authors for shortening the text. The language seems to be improved though the text could still be shortened in a few parts because it contains information that are not related to the topic (examples are lines 69 - 75; lines 81 - 83 provide information from aviculture literature that do not do justice to this topic).

Experimental design

In tables 2 and 4 the number of examined specimens is missing (especially separated in males and females in table 4)

I have never seen a paper in which a new taxon was selectively described as a species or subspecies. Being no expert in taxonomy, I would recommend to the authors to check again exactly whether this is allowed according to the ICZN rules.

Validity of the findings

I would like to congratulate the authors for describing a new Amazona species but, unfortunately, in their revision the authors still left open a number of important questions.

The discussion if the new taxon might be an atypical form of A. albifrons did not convince me because the authors reduce it to a possible extent of intraspecific variation in A.albifrons, and did not discuss the possibility of an aberrant form, which are not unusual in parrots.

As before the main problem is that the dates from the field and origin of the new taxon are still dependent on the parrot owner's statements. I cannot see evidence that one of the authors (which one?) has collected additional field data as mentioned. Though in addition, some new information are added, but they are as blurred as the previous ones. One of several examples is: "Moreover, they do not occupy the same habitat and live commensally. Field studies carried out by Miguel A. Gomez Garza and others during the past 30 years have revealed no hybrids between the new taxon and A. albifrons". This indicates that the new taxon has been known for 30 years and the bird owner has studying the wild population during this period? This information is missing in the first version. I do not want to insinuate Sr. Miguel A. Gomez Garza, that he is giving intentional incorrect information, but I am afraid that the ones provided by him are quite subjective and need to be confirmed or corrected by an experienced and objective field observer.

Additional comments

Taking into account the aforementioned concerns, it is difficult to understand why the authors use their time to deepen their theoretical justification on such an important subject as describing a new species of Amazona, instead of simply going to the field and gathering their data and evidence there. This immediately would erase all doubts.

---

## Round 0.2 · Major Revisions

Dear authors
we reviewed your revision using a new reviewer. As you see, more issues came up which require a major revision.
Greetings
Michael Wink
Academic editor

·

Basic reporting

The English used is mostly good, although a few passages could be improved, e.g. "Amazona genus" should be changed to genus Amazona.

Experimental design

Under 2.2. you listed body weight as one of the 5 metric features. It is difficult to compare between taxa as it depends on such variables as season, physiological status, age, sex, dominance status. availability of food, and so on. Where did you get the weights of the museum specimens from, and are they really comparable to the weights of the two birds of the new taxon?

You did not mention in which museums you looked for possibly misidentified specimens of the new taxon that could have been used as type specimens. It is clearly of interest whether your recherches included the most important ornithological collections world wide, especially those holding significant numbers of Mexican birds.

In figs, 12 & 14, the scietific name of one taxon is lilacina, not lilacine.

Validity of the findings

The "significant differences in morphometry, morphology, behavior and vocal features imply that the new parrat can obtain species status ,,,", and according to your repeated notion that it is sympatric with its closest genetic relative, A. albifrons nana, leave no doubt that it should be treated as a distinctive species according to any species concept!
The relative small genetic differences should not be over-rated. There are many examples where taxa with minor genetic differences are treated as valid species: Apus apus/A. pallidus; Clanga clanga/C. pomarina; Falco rusticolus/F. biarmicus/F. cherrug. Genetic data tell us something about relationships of taxa, but not about its taxonomic rank. That must be deduced from other traits.

I have at hand all descriptions of new species-level taxa of the last 50 years and a lot of older ones (several thousand). There is no case in which a new taxon is proposed as a new species as well as a new subspecies at the same time. Each description of a new taxon is a hypothesis that has to be and will be tested. You should come to a decision: is the new Amazona a new species or a new subspecies? But it cannot be both. Downgrading or upgrading in taxonomic rank is taxonomists' everyday's work, and your influence on it is limited.

It is not quite clear if the living male bird or several of its feathers deposited in the Univ. Nuevo Leon are the holotype (e.g. Lines 517-520)! According to Art. 72.5.1. of the Code, an animal or parts of an animal can be used as type specimens. You should avoid confusing notes and clearly designate one of the two captive birds as the holotype. There is, for example, some confusion about the actual type of the antpitta Grallaria fenwickorum, feathers deposited in the museum, or the living bird depicted on photos. This may have nomenclatural consequences.

Additional comments

Discoveries and descriptions of new taxa are always interesting, in particular those of larger bird species like parrots. Your paper is a welcome contribution to the understanding of biodiversity. However, I recommand that you should clarify your taxonomic statements and that you unequivocally fix the taxonomic status of the new taxon (species or subspecies) and clearly designate the holotype of the new taxon.

Your contribution to the phylogeny and evolution of the genus Amazona is also of much interest.

---

## Round 0.3 · Minor Revisions

Dear authors

You are almost through...

Please consult the annotated ms from the reviewer and make final revisions

Kind Regards

MWink
Editor

·

Basic reporting

With minor exceptions (lines 655-677) the English text is unambiguous and easily understood.
All other points (references, structure of the article) are of professional standard throughout.

Experimental design

no comment

Validity of the findings

no comment

Additional comments

I have the impression that (some of) the authors have problems with their taxonomic conclusion that the new taxon they describe should be treated as a species rather than a subspecies. They often use 'can', 'could be', or 'should', but from the evidence they provide, species status for the new parrot seems well founded.

---

## Round 0.4 · accepted · Accept

Good news - your manuscript is OK by now and can be accepted

Thanks for submitting your research to our journal.

Regards

Michael Wink
Academic editor

·

Basic reporting

no comment

Experimental design

no comment

Validity of the findings

no comment

Additional comments

A very interesting and important paper. I think that it is fine as it now is.